# Research on vehicle lateral stability control under low-adhesion road conditions using proximal policy optimization algorithm

Honglei Pang[1☯], He Huang[2☯]*, Yangping Fan[2,3☯], Lei Yao[2☯], Yong Chen[1☯]

1 School of Transportation Engineering, Nanjing Vocational University of Industry Technology, Nanjing, Jiangsu, Peoples R China, 2 School of Automotive and Transportation Engineering, Hefei University of Technology, Hefei, Anhui, Peoples R China, 3 College of Automotive Engineering, Ganzhou Vocational and Technical College, Ganzhou, Jiangxi, Peoples R. China

☯ These authors contributed equally to this work.
* cranehh@hfut.edu.cn

## Abstract

Vehicle lateral stability control under hazardous operating conditions represents a pivotal challenge in intelligent driving active safety. To address the issue of maintaining vehicle stability during emergency braking on roads with low and non-uniform adhesion, this paper proposes an intelligent integrated longitudinal and lateral stability control algorithm based on the Proximal Policy Optimization (PPO) algorithm. Firstly, high-fidelity models of electromechanical braking (EMB) and steer-by-wire (SBW) systems are constructed in Amesim by leveraging their dynamic characteristics, while a full-vehicle dynamics model is developed in CarSim. The dynamic accuracy of the drive-by-wire system is verified through input-output tracking analysis. Next, vehicle stability is analyzed using vehicle dynamics models to optimize reinforcement learning control variables. This involves designing a continuous state space and action space that incorporate vehicle states and road surface parameters. A multi-objective reward function is formulated using stability indicators, including critical tire slip angle, critical sideslip angle, and critical yaw rate thresholds. Training is conducted via an Amesim-CarSim-Python co-simulation platform for emergency braking scenarios on split-μ roads, low-adhesion surfaces, and curved roads. Results show that, compared to Model Predictive Control (MPC) and Sliding Mode Control (SMC), the PPO algorithm reduces braking distance by 15–20% on low-adhesion roads, decreases lateral deviation by 25–30% on split-μ roads, and suppresses yaw rate oscillation by 28.8% on curved roads. Hardware-in-the-loop (HIL) validation confirms the algorithm's robustness under extreme conditions, with lateral stability metrics maintained within safety thresholds.

## 1. Introduction

Vehicle lateral stability control is a critical technology for preventing accidents under extreme conditions such as sideslip and fishtailing. However, relying solely on

**Data availability statement:** All relevant data are within the paper and its Supporting information files.

**Funding:** This study was supported by the New Talented Researchers of Nanjing Vocational University of Industry Technology (grant No. YK-22-04-03: H. Pang), and the Yangtze River Delta Sci-Tech Innovation Community Joint Research Project (grant No. 2023CSJGG1600: H. Huang).

**Competing interests:** The authors have declared that no competing interests exist.

braking systems or active front steering to balance yaw moments during unstable vehicle states proves insufficient under hazardous conditions. Ensuring stable vehicle motion requires rational and optimal distribution of braking force to each wheel, which maximizes road adhesion utilization while balancing steering stability and braking efficiency. Consequently, integrated longitudinal-lateral stability control coordinating multiple vehicle systems has emerged as a research focus. Zhang et al. [1] proposed a method combining differential braking with active front steering to improve lateral stability in multi-axle vehicles. Mahmoud et al. [2] developed Integrated Torque Vectoring (ITV), which uses yaw rate and vehicle sideslip angle for steering correction in all-wheel-drive electric vehichles. Liang et al. [3] introduced an integrated control framework incorporating torque vectoring (TV) and an active front-wheel steering system (AFS). Their approach employed game theory to design a distributed parallel control strategy for multi-agent coordination and adopted robust H∞ compensation to suppress system disturbances. Experimental results demonstrated that this integrated control framework significantly improves vehicle handling stability.

Research on vehicle stability control algorithms has evolved significantly. Bruni et al. [4] optimized the coordination of Active Front Steering (AFS) and Electronic Stability Program (ESP) using genetic algorithms. Choi [5] implemented Proportional-Derivative (PD) control with range-rate observers to achieve precise lateral tracking. Wang et al. [6] designed an integrated control architecture based on Model Predictive Control (MPC). Zhao et al. [7] incorporated vertical dynamics to enable suspension-integrated stability control. Huang et al. [8] combined LQR trajectory tracking, PID speed control, and direct yaw moment control to enhance vehicle stability. Wang et al. [9] proposed Adaptive Sliding Mode Control (ASMC) with Kalman filtering for parameter self-tuning on complex roads. Despite these advances, traditional methods such as PID, MPC and SMC [10] exhibit considerable performance limitations. These approaches heavily rely on accurate dynamic models, are sensitive to nonlinear tire characteristics, and remain vulnerable to time-varying road parameters. While many studies employ linear vehicle dynamics models for controller design [11,12], these often fail under extreme maneuvering conditions, prompting increased research into nonlinear control techniques [13]. Reference [14] established a Fiala model to characterize tire lateral forces and developed a nonlinear MPC strategy with a fast numerical solver. Experimental results confirmed its effectiveness in enhancing vehicle stability on low-friction roads. Considering the prioritization of control between vehicle handling performance and stability control systems with varying stability margins, the vehicle lateral stability control integrated with safety region boundaries has been extensively studied, which is known as envelope control. Liang et al. [15] addressed tire nonlinearity through fuzzy modeling and proposed a robust H∞ state-feedback controller based on stability margins defined in the tire slip angle phase plane. This controller demonstrated capability in maintaining vehicle performance across varying stability margins.

As research advances, machine learning-based methods have shown promising control performance in handling systems with significant dynamic uncertainties [16,17]. In the field of vehicle dynamics control, addressing model dependency in conventional algorithms and dynamic characteristics of drive-by-wire systems, deep

reinforcement learning (DRL) has shown significant potential for complex nonlinear control tasks through data-driven policy optimization. It effectively addresses issues such as the model dependency of conventional algorithms and the dynamic characteristics of drive-by-wire systems. Abdelhafid et al. [18] proposed a deep neural network (DNN)-based controller capable of correcting lateral and angular deviations caused by trajectory curvature or emergency scenarios, such as sudden braking or obstacle avoidance. Qin et al. [19] developed coordinated control models for longitudinal acceleration and lateral steering angle coordinated control models using Deep Deterministic Policy Gradient (DDPG) and Multi-Agent DDPG (MADDPG), which enhanced both driving safety and comfort. Zhang et al. [20] introduced a novel lane-changing model that integrates longitudinal and lateral control via DDPG, enabling safer and more efficient lane changes with reduced time and distance gaps. Liu et al. [21] combined DDPG for longitudinal speed control with DQN for lateral motion regulation. Aiswarya et al. [22] designed DDPG-based reward functions for integrated lateral-longitudinal control in dynamic environments. However, DDPG is known to be sensitive to hyperparameters and can exhibit tendencies toward overfitting and training instability [23]. Mahmoud et al. [24] demonstrated that a PID controller tuned with Twin Delayed DDPG (TD3) outperformed standard DDPG in lateral stability control, providing smoother control outputs and faster convergence. Alternative policy optimization algorithms, such as Trust Region Policy Optimization (TRPO), Asynchronous Advantage Actor-Critic (A3C), and PPO, offer improved training stability and efficiency [25–27]. Ye et al. [28] implemented a PPO-based lane-changing strategy that achieved success rates of 95% to 99%, benefiting from its strengths in handling continuous actions, flexibility, and broad exploration capability. Xu et al. [29] proposed a deep maximum entropy proximal policy optimization algorithm that incorporates dynamic environment perception, heuristic entropy regularization, and adaptive constraints, this approach achieved rapid convergence and a balance among efficiency, energy consumption, smoothness, and safety through multi-objective reward design and iterative training-testing cycles. Nevertheless, PPO also has drawbacks: it fails to fully incorporate prior knowledge of vehicle dynamics, which can lead to policy degradation in rapidly changing scenarios, slowed convergence in later training stages, uncontrollable training costs, and difficulty in balancing exploration and exploitation in complex environments. These issues often result in reduced smoothness of control actions during rapid attitude adjustments, introducing potential safety risks [30].

To the best of our knowledge, few studies have investigated the application of DRL integrated with longitudinal-lateral coordinated control to improve vehicle stability during emergency braking under hazardous conditions. These conditions are characterized by low-adhesion and unevenly distributed road surfaces. This study pioneers the integration of reinforcement learning with longitudinal-lateral coordinated control for vehicle stability enhancement under such hazardous conditions (e.g., low-adhesion roads with $\mu = 0.2$ and unevenly distributed surfaces). We propose a PPO-based stability control framework (Fig 1) with three key contributions: 1) High-fidelity co-simulation environment: Input-output hysteresis characteristics of drive-by-wire systems were analyzed through Amesim simulations, and vehicle model parameters in CarSim were calibrated to establish a high-precision co-simulation platform. 2) A control optimization reward is designed. By dynamically solving parameters (e.g., peak slip angle, ideal yaw angular acceleration, and ideal centroid slip angle) of the vehicle in combination with the dynamic model, the learning efficiency, convergence speed, and reliability of the training model are enhanced. 3)Based on the validation through Amesim-CarSim-Python co-simulation and HIL testing, the PPO algorithm demonstrates enhanced stability control effectiveness compared to traditional control algorithms across multiple extreme operating conditions, including split-μ road, low-adhesion road, and curved road. This framework establishes theoretical foundations for intelligent vehicles in complex scenarios.

## 2. Vehicle model establishment

This study investigates lateral stability control under low-adhesion road conditions, utilizing the braking and steering systems as primary control targets. High-fidelity models of the electromechanical brake (EMB) and steer-by-wire (SBW) systems were developed in Amesim (mechanical system simulation platform). Meanwhile, vehicle body and tire dynamics were modeled in CarSim (vehicle dynamics simulation software). As shown in Fig 2, the EMB system consists of a brake

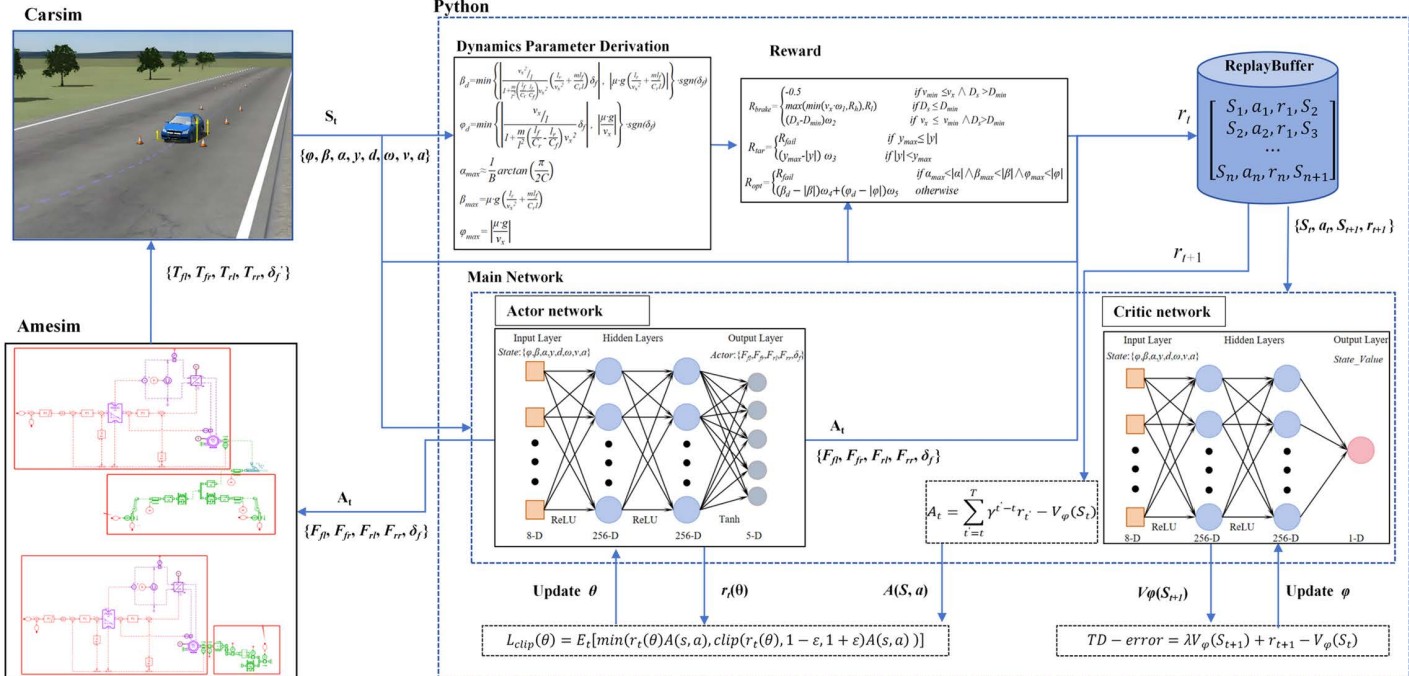

**Fig 1. Overall framework diagram of the study.**

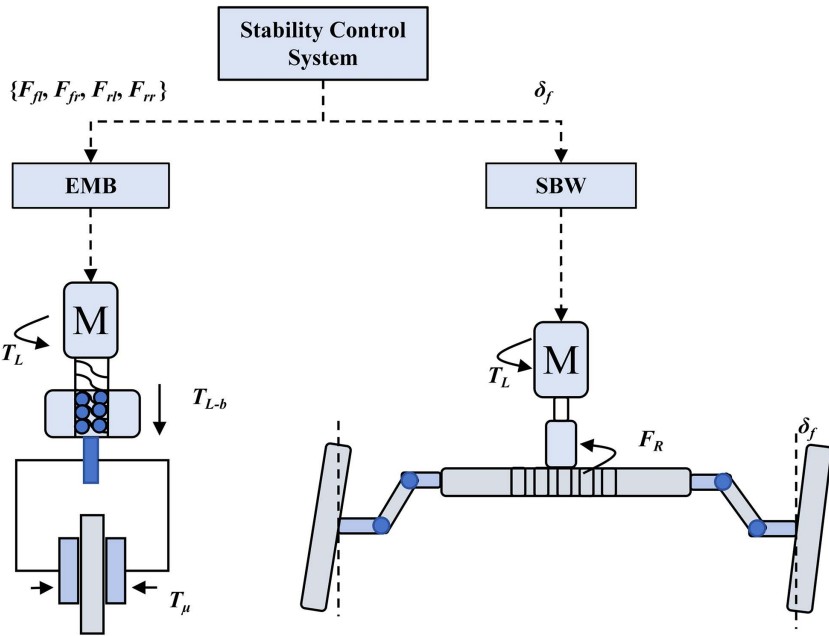

**Fig 2. EMB and SBW system structure.**

motor controller and mechanical transmission assembly. This configuration enables rapid electromagnetic actuation for precise torque delivery. Similarly, the SBW system incorporates a steering motor controller and steering transmission mechanism, achieving high-precision steering torque control.

## 2.1. EMB model

### 2.1.1. Establishment of mechanical structure model for EMB system.

(1) **Brake Motor Model**

Permanent Magnet DC (PMDC) motors are selected as the drive motor for the braking system owing to their low armature inductance. This characteristic enables rapid voltage adjustment and short response times, making PMDC motors suitable for applications that require frequent starts, stops, or speed regulation. The voltage balance equation of the PMDC motor is expressed as:

$$U_a = E_a + I_a R_a + L_a \frac{dI_a}{dt}$$
(1)

where $U_a$ is the armature terminal voltage, $E_a$ is the back electromotive force(EMF), $I_a$ is the armature current, $R_a$ is the armature resistance, $L_a$ is the armature inductance. The back EMF is proportional to the motor speed:

$$E_a = K_e \omega$$
(2)

where $K_e$ is the EMF constant, $\omega$ is the rotor angular velocity.

The mechanical dynamics of the PMDC motor are governed by:

$$J \frac{d_\omega}{d_t} = T_{em} - T_L - B_m \omega$$
(3)

where $J$ is the moment of inertia, $T_{em}$ is the electromagnetic torque, $T_L$ is the load torque, and $B_m$ is the viscous friction coefficient.

The electromagnetic torque $T_{em}$ is proportional to the armature current $I_a$:

$$T_{em} = K_t I_a$$
(4)

where $K_t$ is the torque constant.

(2) **Transmission mechanism model**

The transmission mechanism consists of a planetary gear reducer and a ball screw assembly. The rotation of the motor output shaft drives the ball screw, which converts rotational motion into linear displacement of the nut. The kinematic relationship is described as:

$$S_{bn} = \frac{\theta_b L_b}{2\pi i_t}$$
(5)

where $S_{bn}$ is the horizontal displacement of the nut, $\theta_b$ is the EMB motor rotation angle, $i_t$ is the reducer gear ratio; and $L_b$ is the ball screw lead.

(3) **Clamping Force Model**

The planetary gear reducer amplifies the motor torque and speed, while the ball screw converts torque into caliper pad pressure to generate clamping force. Considering the nonlinear effects of clearance between the ball screw and caliper pads, the clamping force is formulated as:

$$F_n = K_1 \Delta s_{bn}^3 + K_2 \Delta s_{bn}^2 + K_3 \Delta s_{bn} + K_4 \tag{6}$$

where $\Delta S_{bn}$ is the difference between the piston displacement and the brake clearance; $k_1$, $k_2$, $k_3$, $k_4$ are clamping force coefficients.

In addition, when the ball screw structure presses the caliper pad, a reaction force is generated. This force acts on the output shaft of the drive motor through the transmission mechanism, forming a load torque. Its magnitude can be expressed as:

$$T_{L-b} = \frac{F_n}{\left(\frac{2\pi}{L_b}\right) i_t \eta_b \eta_t} \tag{7}$$

where $\eta_b$ is the working efficiency of the ball screw, and $\eta_t$ is the working efficiency of the reducer.

### (4) Brake Disc Model

The piston thrust forces the caliper pads against the brake disc, generating friction torque:

$$T_{\mu'} = 2\mu' F_n r_e \tag{8}$$

where $\mu'$ is the friction coefficient between the pad and disc, and $r_e$ is the effective friction radius of the brake disc.

To address mathematical discontinuities during the transition from static to dynamic friction, a hyperbolic tangent-modified Coulomb friction model is adopted:

$$\mu' = \mu_c tanh\left(2 \times \frac{n_d}{n_c}\right) \tag{9}$$

where $\mu_c$ is the sliding friction coefficient; $n_d$ is the brake disc rotational speed; and $n_c$ is the speed limit.

By adjusting the parameter $\mu_c$, a smooth transition between positive and negative friction torques can be achieved. This approach avoids the issue of numerical mutation at the zero-speed crossing encountered in traditional piecewise linear models. As a result, the numerical stability of brake judder and noise simulations is significantly improved.

**2.1.2. Modeling of EMB system based on Amesim.** Based on the mathematical model derived in Section 2.1.1, the EMB system model built in Amesim does not include tire models, brake caliper models, etc. [31]. The EMB system model and its main parameters are shown in Fig 3 and Table 1, respectively.

## 2.2. SBW model

### 2.2.1. Establishment of mechanical structure model for SBW system.

#### (1) Rack-and-Pinion Model

Kinematic analysis of the pinion and rack yields the governing equation:

$$m_r \ddot{x} + b_r \dot{x} + F_R = \frac{K_m}{r_p}\left(\theta_m - \frac{x}{r_p}\right) \tag{10}$$

where $x$ is the rack-and-pinion displacement, $K_m$ is the rigidity coefficient of the motor output shaft, $m_r$ is the equivalent mass of the pinion and rack, $F_R$ is steering resistance, $b_r$ is the damping coefficient of the rack, $r_p$ is the pinion radius, and $\theta_m$ is the motor torque.

To simplify model complexity, the steering resistance is expressed as:

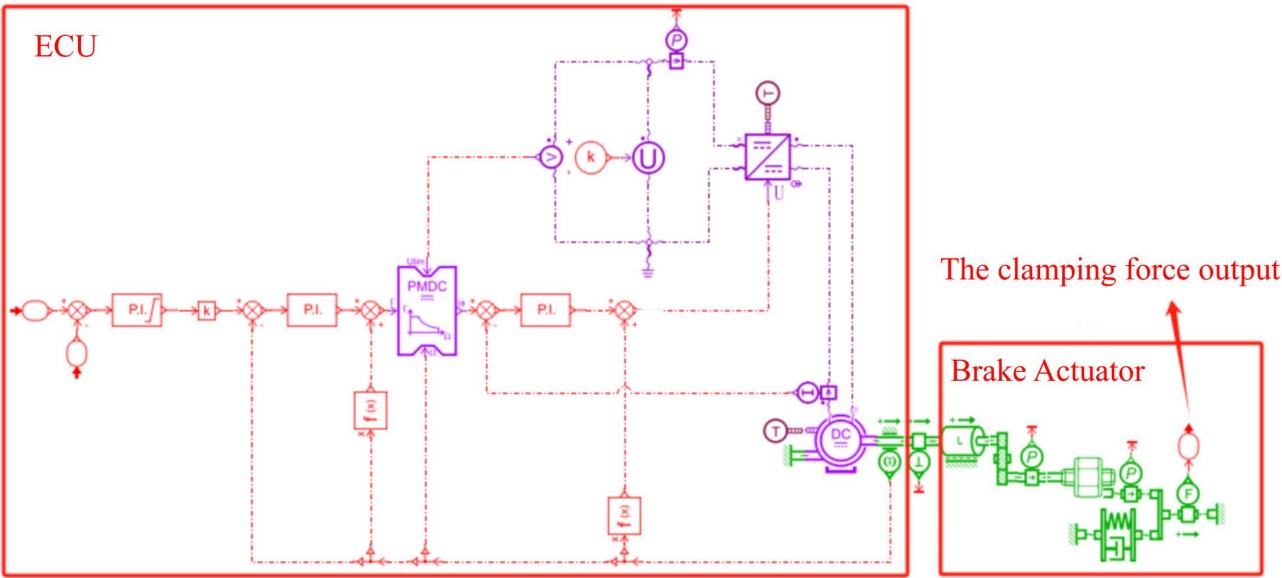

**Fig 3. Braking system model.**

**Table 1. Main parameters of braking system.**

| Parameter | Value | Unit |
|---|---|---|
| Reference temperature | 25 | ° |
| Motor moment of inertia | 0.002 | kg/m² |
| Motor torque coefficient | 0.65 | N·m/A |
| Motor EMF coefficient | 0.193 | v·s/rad |
| Armature resistance | 0.56 | Ω |
| Armature inductance | 0.3 | mH |
| Motor correction coefficient | 0.01 | Null |
| Reduction mechanism transmission ratio | 1:20 | Null |
| Ball screw lead | 12 | mm |

$$F_R = K_r x + F_d \qquad (11)$$

where ke is the elastic coefficient of the equivalent spring, and $F_d$ denotes road excitation signals.

(2)    Front-Wheel Steering Angle Model

The front-wheel steering angle is determined by the displacement relationship between the steering linkage and wheel rotation, governed by:

$$J_f\ddot{\theta}_f + C_{kf}\dot{\theta}_f + M_z = K_{rp}\left(\frac{x}{R_{st}} - \theta_f\right) \qquad (12)$$

where $J_f$ is the moment of inertia of the front wheel about the kingpin, $M_z$ is the self-aligning torque of the steering system, $C_{kf}$ and $K_{rp}$ are the damping and stiffness coefficients of the kingpin, respectively, and $R_{st}$ is the transmission ratio from the rack to the steering wheel.

**2.2.2. Modeling of SBW system based on Amesim.** Based on the mathematical model derived in Section 2.3.1, the SBW model is built in Amesim. The steer-by-wire system model and its main parameters are shown in Fig 4 and Table 2, respectively.

## 2.3. Modeling of vehicle body and environment models based on CarSim

The vehicle model is implemented in CarSim with a simplified Magic Formula tire model [32] to characterize lateral force dynamics, while the simulation scenarios are constructed based on the requirements of control algorithm validation. The fundamental vehicle body parameters are summarized in Table 3, and 225/60 R18 tires are adopted for modeling.

**2.3.1. Design of simulation scenarios.** To enhance the control robustness of conventional algorithms under hazardous conditions, three typical accident-prone scenarios are implemented in CarSim: curved road, low-friction road, and split-μ road. Each scenario is parameterized with road geometry and curvature, vehicle placement, obstacle vehicles, reference trajectory, initial conditions, friction coefficients, and other relevant variables.

The split-μ road surface and low-adhesion road surface simulation scenarios are configured as a 200 m straight single-lane road. For the split-μ condition, the adhesion coefficient is set to 0.4 on the left side and 0.8 on the right side. The low-adhesion scenario uses a uniform coefficient of 0.2. The curved road scenario with a radius of 100 m and a length of approximately 628 m is designed with a bilateral adhesion coefficient of 0.8 to evaluate lateral stability. In all

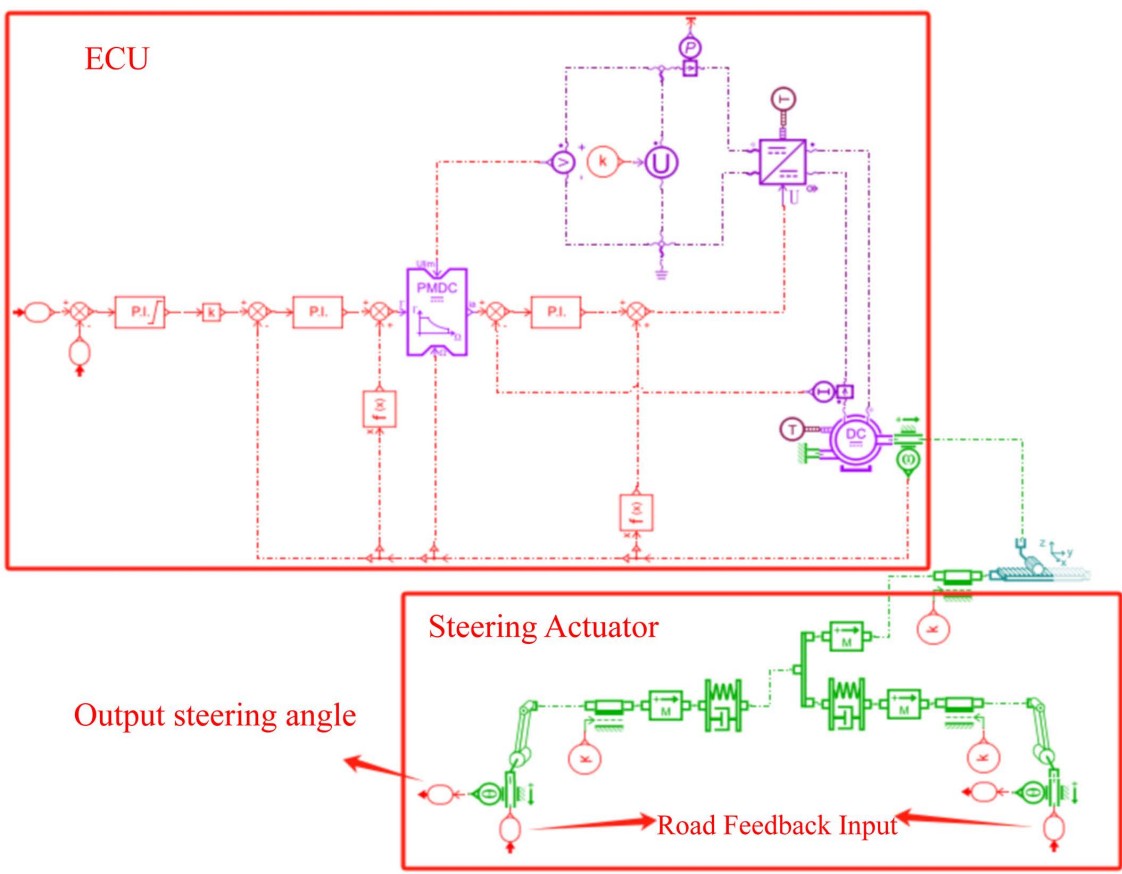

**Fig 4. Steering system model.**

**Table 2. Main parameters of steering system.**

| Parameter | Value | Unit |
|---|---|---|
| Reference temperature | 25 | ° |
| Motor moment of inertia | 0.06 | kg/m² |
| Motor damping coefficient | 0.5 | N·m·s/rad |
| Armature inductance | 5 | mH |
| Motor EMF coefficient | 0.078 | v·s/rad |
| Motor correction coefficient | 0.65 | Null |
| Motor torque coefficient | 0.65 | N·m/A |
| Armature resistance | 0.26 | Ω |
| Pinion radius of rack-and-pinion | 7 | mm |

**Table 3. Vehicle parameters.**

| Parameter | Value | Unit |
|---|---|---|
| Total vehicle mass | 1270 | kg |
| Height of center of mass | 540 | mm |
| Wheelbase | 2910 | mm |
| Distance from center of mass to front axle | 1015 | mm |
| Distance from center of mass to rear axle | 1895 | mm |
| Moment of inertia about the X-axis | 536.6 | kg·m² |
| Moment of inertia about the Z-axis | 1536.7 | kg·m² |

three scenarios, a stationary obstacle vehicle is placed at the 100 m mark. This setup is used to evaluate the vehicle's braking response and calculate the braking distance once the obstacle is detected.

**2.3.2. Input-output parameter configuration.** The input variables of the simulation environment are set as: braking torque of the four wheels and front wheel steering angle; the output variables include lateral displacement, vehicle sideslip angle, yaw rate, yaw angular acceleration, braking distance, wheel slip ratio, vehicle velocity, and braking deceleration. The scenario simulation step size is set to 0.01s, and the maximum simulation time is 10s.

## 2.4. Establishment of simulation model response verification

To validate the correctness of the established model, a co-simulation framework was implemented by integrating the EMB and SBW systems developed in Amesim with the full-vehicle model in CarSim. Control commands were injected via Simulink to evaluate the response characteristics of steering and braking systems. Step signals and sinusoidal signals are input to the brake system and steering system, respectively.

### (1) EMB system

Step Input: A clamping force control signal of 1,500N was applied 2s after the simulation initiation. Sinusoidal Input: A sinusoidal clamping force signal with an amplitude of 1,500N and a frequency of 0.5 Hz was injected at the simulation startup. Simulation validation assessed the system's transient and steady-state performance under two clamping force excitation profiles (Fig 5): 1) Step Input (applied at 2s): Rise time≈0.1s; Peak force≈1,548N (overshoot≈3.2%); Settling time≈0.3s. 2) Sinusoidal Input (0.5 Hz, 1,500N amplitude): Steady-state error<1.92%; Settling time≈0.5s. The results confirm the braking system's dual capabilities: Rapid transient adaptation (settling under 0.3s) for dynamic command tracking, and high force fidelity (steady-state error below 2%) under periodic excitation. This performance profile meets the upper-layer control requirements for both fast actuation and precise force regulation in hazardous scenarios.

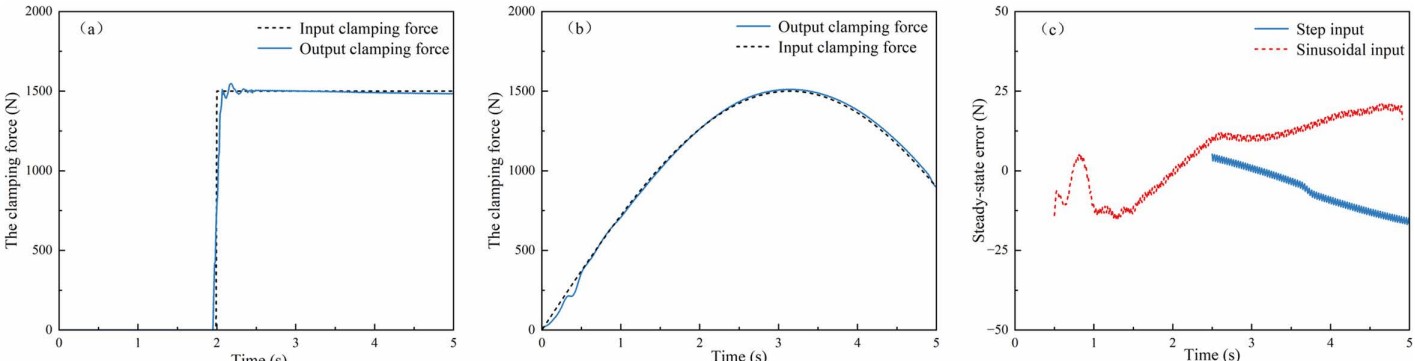

**Fig 5. Simulation Results Plot of the EMB System Response: (a) Step response curve of the EMB system; (b) Sine response curve of the EMB system; (c) Steady state error curve of the EMB system.**

### (2)  SBW system

Step Input: A wheel steering angle control signal of 10° was applied 1.8 s after simulation initiation. Sinusoidal Input: A sinusoidal wheel steering angle signal with an amplitude of 40° and a frequency of 0.5 Hz was injected at simulation startup. Simulation validation assessed the transient and steady-state performance under two steering angle excitation profiles (Fig 6): 1) Step Input (applied at 1.8 s): Rise time≈0.2 s; Peak angle≈10.1° (negligible overshoot); Settling time≈0.3s. 2) Sinusoidal Input (0.5 Hz, 40° amplitude): Settling time≈0.05s; Steady-state error<4.68% of target value. The results confirm the steering system's dual capabilities: Rapid transient adaptation (settling under 0.3s) for dynamic command tracking; and high angle fidelity (steady-state error below 5%) under periodic excitation.This performance profile satisfies the upper-layer control requirements for both fast actuation and precise angle regulation.

## 3.  Longitudinal-lateral stability control algorithm design based on PPO

### 3.1.  Vehicle stability control analysis

Traditional control algorithms, such as MPC and SMC, utilize desired braking deceleration and target yaw rate as reference variables. Constraint thresholds are applied to the yaw rate, tire slip angle, and combined longitudinal-lateral tire

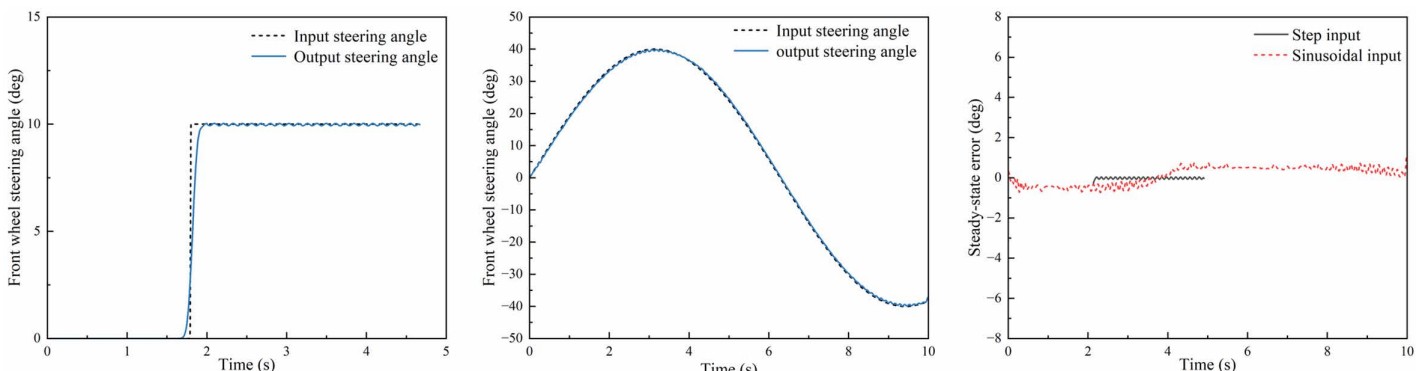

**Fig 6. Simulation Results Plot of the SBW System Response: (a) Step response curve of the SBW system; (b) Sine response curve of the SBW system; (c) Steady state error curve of the SBW system.**

forces to ensure both control effectiveness and adaptability across diverse operating conditions. In contrast, DRL model training relies on the evaluation of vehicle stability metrics. Predefined stability thresholds are established, beyond which the vehicle is considered unstable. Additionally, rate limitations are enforced on control variables (such as braking torque and steering angle) to prevent abrupt actuator actions that could compromise system stability.

**3.1.1. Vehicle instability state judgment.** Identifying the root causes of potential instability under various scenarios is essential for designing effective vehicle stability control methods [33]. In vehicle dynamics, the fundamental mechanism of instability stems from the trade-off between lateral and longitudinal tire forces, as constrained by the friction ellipse theory. The lateral force increases with the slip angle within a certain threshold. Beyond this threshold, however, lateral force saturation occurs, resulting in a sharp decline in tire grip and further aggravating instability. Therefore, it is crucial to maintain the slip angle within allowable limits to ensure stable vehicle motion [34–36]. Consequently, this study adopts maximum slip angle thresholds as one of the instability criteria. The conditions for instability judgment are defined as follows:

$$\begin{cases} \left| \dot{\beta} + \varepsilon_1 \beta \right| \leq \varepsilon_2 \\ \left| \Delta\varphi \right| \leq \left| \varepsilon_3 \varphi_d \right| \\ \left| \alpha \right| \leq \left| \alpha_{max} \right| \end{cases} \tag{13}$$

Where $\varepsilon_1$, $\varepsilon_2$ are adjustable parameters, $\varepsilon_3 = 0.005$ is the yaw rate control coefficient [37], $\beta$ and $\dot{\beta}$ represent the vehicle sideslip angle and vehicle sideslip angle rate; $\varphi$ and $\varphi_d$ denote the yaw rate and ideal yaw rate; $\alpha$ and $\alpha_{max}$ correspond to the tire slip angle and its critical threshold.

According to reference [38], the calculation formulas for $\varepsilon_1$ and $\varepsilon_2$ are presented in Equation (14).

$$\begin{cases} \varepsilon_1 = 0.783\mu^2 - 3.793\mu - 0.632 \\ \varepsilon_2 = 0.079\mu^2 + 0.147\mu + 0.033 \end{cases} \tag{14}$$

While the yaw rate and sideslip angle increase along with the ideal yaw rate, excessively high ideal values under extreme conditions signal an impending loss of control. Therefore, a violation of any of the three stability criteria presented in Equation (13) suggests that the vehicle is in or approaching an unstable state.

**3.1.2. Dynamics parameter derivation.** To guide the vehicle in maintaining stability within the DRL model, this study employs three metrics for evaluating lateral stability: the deviation between desired and actual yaw rates, the deviation between desired and actual vehicle sideslip angles, and the vehicle lateral displacement. To improve DRL training efficiency, critical reference variables that characterize complete instability are defined to prevent ineffective training. These variables include the critical tire slip angle($\alpha_{max}$), critical vehicle sideslip angle($\beta_{max}$), ideal vehicle sideslip angle($\beta_d$), critical yaw angle($\varphi_{max}$), and ideal yaw rate($\varphi_d$), all of which are derived from the dynamic model. Such thresholds enable the DRL agent to learn a balance between stability and control efficiency while adhering to vehicle dynamics constraints. By integrating these constraints into the reward function, the agent is guided to prioritize stability-critical behaviors without compromising computational efficiency. The proposed framework ensures compliance with physical feasibility limits, thereby enhancing convergence stability during policy optimization.

**(1) Critical tire slip angle**

Derived from the tire magic formula model, the critical tire slip angle is calculated by assuming the tire lateral force reaches its peak value.

$$\alpha_{max} \approx \frac{1}{B} arctan\left(\frac{\pi}{2C}\right) \tag{15}$$

Where B represents the lateral shape factor, and C denotes the stiffness factor.

 

**(2) Critical vehicle sideslip angle and ideal vehicle sideslip angle**

Lateral stability control requires consideration of road adhesion constraints and vehicle dynamic characteristics to determine $\varphi_{max}$ and $\varphi_d$. Based on the 2-degree-of-freedom (2-DOF) vehicle model:

$$\begin{cases} (C_f+C_r)\frac{v_y}{v_x}+\frac{1}{v_x}(C_fl_f-C_rl_r)\varphi-C_f\delta_f=m(\dot{v}_y+v_x\varphi) \\ (C_fl_f-C_rl_r)\frac{v_y}{v_x}+\frac{1}{v_x}\left(C_fl_f^2+C_rl_r^2\right)\varphi-l_fC_f\delta_f=I_z\dot{\varphi} \end{cases} \tag{16}$$

Where $m$ denotes the total vehicle mass; $C_f$ and $C_r$ represent the front and rear tire cornering stiffness, respectively; $v_x$ represents the longitudinal velocity; $\delta_f$ is the front wheel steering angle; $v_y$ refers to the lateral velocity; $I_z$ stands for the yaw moment of inertia about the vertical axis; $\dot{\varphi}$ is the yaw angular acceleration; $l_f$ and $l_r$ denote the distances from the center of mass to the front and rear ax/es, respective/y.

The ideal yaw rate plays a pivotal role in ensuring that vehicle steering responsiveness aligns with control expectations. It prevents excessive sensitivity or lag, thereby providing a precise and reliable control target for control algorithms. This derivation assumes the vehicle operates under steady-state driving conditions, where the yaw angular acceleration and lateral acceleration are negligible (i.e., $\dot{\varphi}=0$; $\dot{v}_y=0$). Substituting these conditions into the Equation (16) yields the following two-degree-of-freedom (2-DOF) dynamic model for steady-state motion:

$$\begin{cases} (C_f+C_r)\frac{v_y}{v_x}+\frac{1}{v_x}(C_fl_f-C_rl_r)\varphi-C_f\delta_f=mv_x\varphi \\ (C_fl_f-C_rl_r)\frac{v_y}{v_x}+\frac{1}{v_x}\left(C_fl_f^2-C_rl_r^2\right)\varphi-l_fC_f\delta_f=0 \end{cases} \tag{17}$$

Following from Equation (16):

$$\varphi=\frac{\frac{v_x}{l}}{1+\frac{m}{l^2}\left(\frac{l_f}{C_r}-\frac{l_r}{C_f}\right)v_x^2}\delta_f \tag{18}$$

However, the lateral acceleration is constrained by the road-tire adhesion coefficient ($\mu$), and under non-ideal driving conditions where vehicle dynamics deviate from theoretical assumptions, the yaw rate ($\varphi$) is ultimately limited by the physical adhesion boundary. This yields the relationship: $\varphi v_x \le \mu g$. Consequently, it follows that:

$$|\varphi|\le\left|\frac{\mu\cdot g}{v_x}\right| \tag{19}$$

Therefore, the $\varphi_{max}$ and $\varphi_d$ can be formulated as:

$$\varphi_{max}=\left|\frac{\mu\cdot g}{v_x}\right| \tag{20}$$

$$\varphi_d=min\left\{\left|\frac{\frac{v_x}{l}}{1+\frac{m}{l^2}\left(\frac{l_f}{C_r}-\frac{l_r}{C_f}\right)v_x^2}\delta_f\right|,\left|\frac{\mu\cdot g}{v_x}\right|\right\}\cdot sgn(\delta_f) \tag{21}$$

**(3) Critical yaw angle and ideal yaw rate**

When the vehicle sideslip angle is small, it can be simplified as: $\beta\approx arctan\frac{v_y}{v_x}\approx\frac{v_y}{v_x}$, From Equation (16), it follows that:

$$\begin{cases} (C_f+C_r)\beta+\frac{1}{v_x}(C_fl_f-C_rl_r)\varphi-C_f\delta_f=mv_x\varphi \\ (C_fl_f-C_rl_r)\beta+\frac{1}{v_x}\left(C_fl_f^2+C_rl_r^2\right)\varphi-l_fC_f\delta_f=0 \end{cases} \tag{22}$$

Following from Equation (21):

$$\beta=\frac{\frac{v_x^2}{l}}{1+\frac{m}{l^2}(\frac{l_f}{C_r}-\frac{l_r}{C_f})v_x^2}(\frac{l_r}{v_x^2}+\frac{ml_f}{C_rl})\delta_f=\varphi v_x(\frac{l_r}{v_x^2}+\frac{ml_f}{C_rl})$$

(23)

Considering the constraints imposed by the road adhesion coefficient, the following equation can be derived:

$$|\beta|\le\left|\mu\cdot g\left(\frac{l_r}{v_x^2}+\frac{ml_f}{C_rl}\right)\right|$$

(24)

Therefore, the $\beta_{max}$ and $\beta_d$ can be formulated as:

$$\beta_{max}=\mu\cdot g\left(\frac{l_r}{v_x^2}+\frac{ml_f}{C_rl}\right)$$

(25)

$$\beta_d=min\left\{\left|\frac{\frac{v_x^2}{l}}{1+\frac{m}{l^2}\left(\frac{l_f}{C_r}-\frac{l_r}{C_f}\right)v_x^2}\left(\frac{l_r}{v_x^2}+\frac{ml_f}{C_rl}\right)\delta_f\right|,\ \left|\mu\cdot g\left(\frac{l_r}{v_x^2}+\frac{ml_f}{C_rl}\right)\right|\right\}\cdot sgn(\delta_f)$$

(26)

## 3.2. Proximal Policy Optimization (PPO) algorithm

The Proximal Policy Optimization (PPO) algorithm was introduced by Schulman et al. in 2017 [27]. In PPO, the update magnitude of the policy is constrained by regulating the ratio between the new and old policies, which is clipped within the range$(1-\varepsilon,\ 1+\varepsilon)$, where $\varepsilon$ is a hyperparameter typically set to $\varepsilon=0.2$. The clipped objective loss function is formulated as:

$$L_{clip}(\theta)=E_t\left[min\left(r_t(\theta)\hat{A}_t, clip(r_t(\theta),1-\varepsilon,1+\varepsilon)\hat{A}_t\right)\right]$$

(27)

where $r_t(\theta)=\frac{\pi_\theta(a_t|S_t)}{\pi_{old\theta}(a_t|S_t)}$ represents the probability ratio between the new and old policies for selecting action $a_t$ at state $S_t$, $\hat{A}_t$ denotes the advantage estimate function, quantifying the disparity between predicted rewards and state values. The advantage function is typically computed as:

$$A_t=\sum_{t'=t}^{T}\gamma^{t'-t}r_{t'}-V_\varphi(S_t)$$

(28)

where $V_\varphi(S_t)$ denotes the state value estimated by the Critic network, $r_{t'}$ is the immediate reward at step $t'$, and $\gamma\in[0,1]$ is the discount factor.

**3.2.1. Markov decision process modeling.** This study focuses on vehicle stability control under hazardous braking conditions. The stability control problem in extreme scenarios is formulated as a Markov Decision Process (MDP), which effectively represents vehicle states, actions, and policy performance.

(1) **State Space**

The objective is to maintain lateral stability and trajectory tracking during braking under extreme conditions, while minimizing braking distance loss. The state space is defined using three categories of variables: Stability metrics: yaw rate ($\varphi$), vehicle sideslip angle ($\beta$), and tire slip angle ($\alpha$); Trajectory tracking indicators: lateral displacement ($y$); Braking efficiency parameters: braking distance ($d$), wheel slip ratio ($\omega$), vehicle speed ($v$), and braking deceleration ($a$).This state representation enables the MDP framework to holistically evaluate vehicle dynamics, trajectory adherence, and braking performance under extreme conditions.The state space is formulated as:

$$State = \{\varphi, \beta, \alpha, y, d, \omega, v, a\} \tag{29}$$

**(2)  Action Space**

The braking and steering systems are modeled in Amesim, while the vehicle body model and simulation environment are constructed in CarSim. During training, the algorithm first outputs the action space as commands applied to the braking calipers (clamping force) and steering angles (control signals). These commands are transmitted to the braking and steering systems through the electronic control module in Amesim, which when adjusts the sub-motors to actuate the mechanical components. Subsequently, the clamping force applied to the braking calipers is converted into wheel braking torque based on the full-vehicle model parameters. This braking torque is combined with the wheel steering angle to control the agent in CarSim. The action space is defined as the braking forces and steering angles applied to each wheel:

$$Actor = \{F_{fl}, F_{fr}, F_{rl}, F_{rr}, \delta_f\} \tag{30}$$

where $F_{fl}$, $F_{fr}$, $F_{rl}$ and $F_{rr}$ denote the braking forces on the left-front, right-front, left-rear, and right-rear wheels, respectively, and $\delta_f$ represents the front wheel steering angle.

**(3)  Network Structure**

The neural network architecture is a critical determinant of model performance. As the state inputs in this study lack high-dimensional data (e.g., images), convolutional operations are dispensable. Fully connected (FC) layers are employed instead for their efficacy in directly mapping state-action relationships within low-dimensional spaces. Furthermore, to meet the stringent real-time response requirements of vehicle stability control, we constrain the network depth to avoid the computational latency and increased training time associated with excessively wide or deep architectures.

Specifically, the Critic network accepts an eight-dimensional state vector (including yaw rate, sideslip angle, etc.) and processes it through a feedforward network with two hidden layers. The first hidden layer projects the input into a 256-dimensional feature space, applying a ReLU activation function for nonlinear transformation. The second hidden layer maintains this 256-dimensional representation for further feature abstraction via ReLU. A final linear output layer then produces a scalar state-value estimate. The Actor network shares the same state input but outputs a five-dimensional action vector (e.g., left-front wheel braking force). Its structure mirrors that of the Critic: two 256-node hidden layers with ReLU activation. The output layer employs a tanh activation function to compress the features into five dimensions and constrain each output value to the range (−0.5, 0.5), ensuring compatibility with the downstream control system's operational requirements.

**3.2.2.  Reward function design based on dynamic model.**  In reinforcement learning, the reward function serves as the pivotal mechanism for guiding the agent toward specific objectives. Its design critically influences the agent's learning effectiveness, convergence speed, and ultimate policy performance. During emergency braking, the agent must maintain vehicle stability while minimizing braking distance. To prevent the agent from deliberately adopting slow braking strategies to artificially maintain stability, a stationary obstacle vehicle is introduced into the simulation environment. The agent is required to execute deceleration maneuvers upon detecting the obstacle. The distance to the obstacle at the moment of braking initiation is used as a key performance metric. This configuration helps avoid inefficient training episodes caused by unstable braking behavior.

**(1)  Braking Efficiency Reward**

Braking efficiency is defined as reducing longitudinal velocity below a minimum threshold. During braking control, when the agent vehicle's velocity decreases to the minimum threshold $v_{min}$, it is deemed to have successfully stopped.

Conversely, any collision with an obstacle vehicle during braking indicates excessive braking distance, classifying the control as a failure. To optimize training efficiency, the simulation terminates immediately upon success or failure by setting the termination flag done = 0, followed by initiating the next training cycle. For successful stops, rewards are allocated based on actual braking distance; for collisions resulting in simulation failure, penalties are determined by the agent vehicle's velocity magnitude at impact. Critically, a minor penalty per training step is enforced throughout braking to incentivize rapid deceleration and expedite simulation termination. Rewards are allocated based on braking performance:

$$R_{brake} = \begin{cases} -0.5 & \text{if } v_{min} \leq v_x \wedge D_s > D_{min} \\ max\left(min\left(v_x \cdot \omega_1, R_h\right), R_l\right) & \text{if } D_s \leq D_{min} \\ \left(D_s - D_{min}\right)\omega_2 & \text{if } v_x \leq v_{min} \wedge D_s > D_{min} \end{cases} \tag{30}$$

Where $v_{min}$ represents the minimum vehicle speed, $v_x$ stands for the longitudinal vehicle speed, $D_s$ indicates the distance between the centroid of the agent and the obstacle vehicle, and $D_{min}$ is the minimum centroid distance between the agent and the obstacle vehicle without collision, $\omega_1$ and $\omega_2$ are weighting parameters, and $R_l$ is the minimum penalty reward with a value less than 0, and $R_h$ is the maximum penalty reward with a value less than 0.

### (2) Trajectory Tracking Reward

The primary training objective is to ensure precise trajectory tracking and lateral stability (sideslip avoidance and fishtailing mitigation). The reward function is designed as a piecewise mechanism: quadratic decay for lateral displacements exceeding the maximum allowable threshold ($y_{max}$), and constant positive reinforcement within the threshold range.

$$R_{tar} = \begin{cases} R_{fail} & \text{if } y_{max} \leq |y| \\ \left(y_{max} - |y|\right)\omega_3 & \text{if } |y| < y_{max} \end{cases} \tag{31}$$

Where $y_{max}$ is the allowable lateral displacement threshold, $\omega_3$ is a weighting parameter, and $R_{fail}$ (negative value) penalizes stability loss. Training terminates (done = 0) if $|y| > y_{max}$.

### (3) Control Optimization Reward

To enhance learning efficiency, dynamics-derived metrics are integrated (Section 3.1). This includes deviations from ideal sideslip angles ($\beta_d$) and yaw rates ($\varphi_d$), bounded by critical stability thresholds ($\beta_{max}$, $\varphi_{max}$):

$$R_{opt} = \begin{cases} R_{fail} & \text{if } \alpha_{max} < |\alpha| \wedge \beta_{max} < |\beta| \wedge \varphi_{max} < |\varphi| \\ \left(\beta_d - |\beta|\right)\omega_4 + \left(\varphi_d - |\varphi|\right)\omega_5 & \text{otherwise} \end{cases} \tag{32}$$

Where $\omega_4$ and $\omega_5$ are weighting parameters.

### (4) Reward Function Parameters

The reward function values in the PPO algorithm are designed as dimensionless parameter values. The parameter values are listed in Table 4. This hierarchical reward structure balances stability, efficiency, and trajectory adherence across diverse driving scenarios.

**3.2.3. Algorithm hyperparameters.** In the PPO framework, achieving optimal training dynamics requires systematic calibration of internal hyperparameters. This study adopted a two-stage hyperparameter optimization scheme to efficiently identify the optimal configuration. The first stage utilized a coarse-grained Bayesian optimization approach for the high-impact hyperparameters [39]: the clipping range (ε) and the learning rate (η). The value of ε was sampled from the interval [0.1, 0.3] with a step size of 0.02. Concurrently, η was sampled logarithmically across the range [1e-5,

**Table 4. Reward function related parameters.**

| Parameter | Rl | Rh | Rfail | ymax | ω1 | ω2 | ω3 | ω4 | ω5 |
|-----------|-----|------|-------|------|-----|----|----|----|----|
| Value | −50 | −100 | −50 | 0.5 | −2 | 10 | 1 | 10 | 8 |

1e-3], encompassing the typical operating region for deep reinforcement learning algorithms. A total of 50 independent training trials were executed, with the average cumulative reward obtained after 500,000 training steps serving as the performance metric for the Bayesian optimizer. Gaussian kernel density estimation was then applied to the results of these trials to identify the three parameter sets associated with the highest probability density. The optimal combination from this candidate set was subsequently determined and validated through three additional repetitive training runs. In the second stage [40], a fine-grained grid search was conducted for the low-impact hyperparameters, namely the replay buffer length (T) and the batch size (B). With the optimal values of $\varepsilon$ and $\eta$ fixed from the previous stage, a comprehensive parameter grid was constructed with T=[512, 1024, 2048] and B = [64, 128, 256]. A full-factorial experimental design was employed, encompassing all $3 \times 3 = 9$ possible combinations. Each combination was evaluated over three independent training sessions. A composite evaluation metric was devised, combining the average number of steps to convergence (weighted 0.6) and the standard deviation of the final reward (weighted 0.4), thereby quantifying the trade-off between convergence speed and training stability. The parameter set that optimized this metric while ensuring sample diversity was selected as the final configuration.

The classification, search spaces, and optimally identified values for all key hyperparameters are summarized in Table 5, while all other parameters not explicitly mentioned remain unmodified from the default settings of the PPO algorithm.

## 4. Co-simulation analysis

### 4.1. Co-simulation environment setup

This study develops a PPO algorithm model in Python and establishes a co-simulation framework that integrates CarSim and Amesim via API interfaces. As illustrated in the co-simulation architecture (Fig 7), CarSim transmits the vehicle state space (e.g., longitudinal velocity, lateral displacement) to Python. The PPO algorithm then generates the action space, which consists of desired brake pad clamping forces and front wheel steering angles. These commands are transmitted to Amesim. Within Amesim, the commands are processed EMB and SBW systems. The system outputs the actual clamping forces and steering angles, which are then returned to Python. In Python, the clamping forces are converted into braking torques using brake disc and tire models. Finally, the braking torque and steering angle control commands are fed back to CarSim to execute the simulation, thereby closing the control loop.

**Table 5. Hyperparameter Configurations in PPO Training.**

| Category | Parameter | Search ranges | Value |
|----------|-----------|---------------|-------|
| High-impact hyperparameters | Clipping range ($\varepsilon$) | [0.1, 0.3] | 0.2 |
| | Learning rate ($\eta$) | [1e-5, 1e-3] | $3 \times 10^{-3}$ |
| Low-impact hyperparameters | Replay buffer (T) | [512, 1024,2048] | 1024 |
| | Batch size (B) | [64, 128, 256] | 64 |
| Fixed values | Max steps | 500000 | 500000 |
| | Discount factor ($\gamma$) | 0.99 | 0.99 |
| | Random seeds | [0, 10,000] | [0, 10,000] |
| | Training epochs | 10 | 10 |

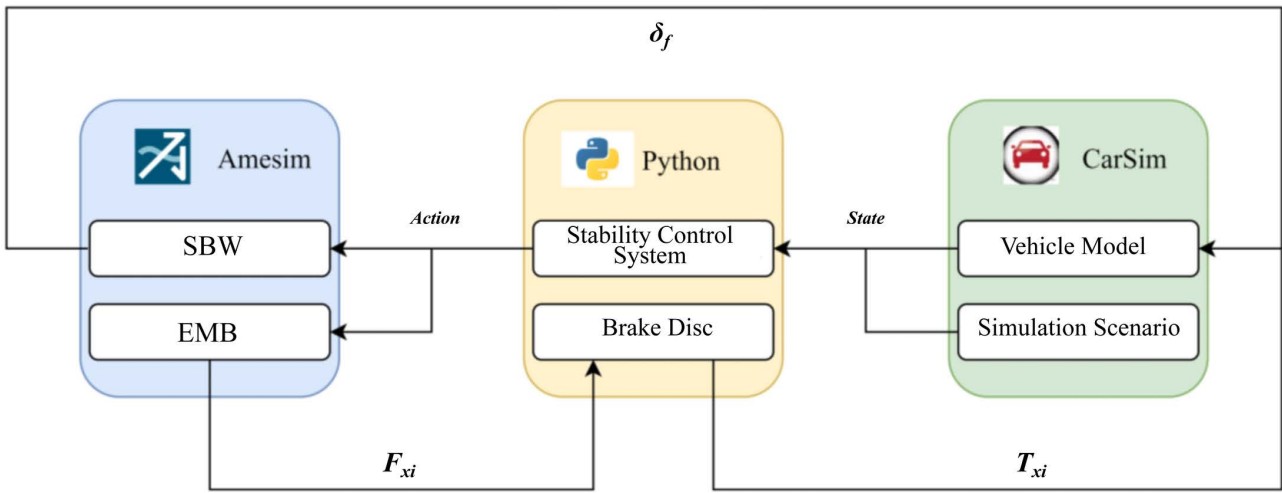

**Fig 7. Co-simulation architecture.**

## 4.2. Comparative analysis of simulation training results

To demonstrate the generalizability of the PPO algorithm for vehicle emergency braking across diverse environments, experiments were conducted on split-μ, low-adhesion, and curved roads to obtain empirical results. To validate the effectiveness and superiority of the PPO-based braking control strategy, a comparative framework (MPC + SMC) was designed, incorporating a MPC-based lateral stability algorithm and a SMC-based braking torque allocation method. Controller models for these baseline algorithms were implemented in Simulink and co-simulated with CarSim to generate comparative experimental data.

**4.2.1. Training convergence analysis.** Training was conducted across three distinct environments. The initial braking speeds were randomized between 50−100 km/h at 10 km/h intervals, in compliance with ISO 14512−1999 and ISO 16234−2006 standards. Training episodes were terminated early if the agent exhibited trajectory deviation, excessive braking distance, or loss of stability. The total number of training steps were set to 500,000 for both split-μ and low-adhesion roads, and 400,000 for the curved road scenario. A fixed timestep of 0.01 s was used throughout all experiments.

The average reward per iteration (Fig 8) illustrates clear convergence characteristics: For the split-μ road scenario, rapid reward growth initiates at 140,000 steps, with stabilization achieved after 350,000 steps. In the low-adhesion road scenario, accelerated reward accumulation commences at 150,000 steps, reaching stabilization post-340,000 steps. For the curved road scenario, early-phase reward acceleration starts at 40000 steps, followed by a gradual growth deceleration after 120,000 steps, and final stabilization at 350,000 steps. These findings validate the efficacy of the reward function design and confirm the PPO algorithm's strong generalizability, as it enables effective control across varying speeds and scenarios.

**4.2.2. Comprehensive training results and comparative analysis.** For each scenario, training covered initial braking speeds ranging from 50 km/h to 100 km/h. This study focuses on representative initial speeds for detailed analysis. Simulations terminated when vehicle speed decreased to ≤0.1 km/h, indicating a full-stop condition.

**(1) Split-μ Road Emergency Braking**

The experiment was configured with an initial velocity of 80 km/h and asymmetric road adhesion coefficients (left: 0.8, right: 0.4) to replicate split-μ conditions. As shown in Fig 9(a, b), the velocity-time profile and braking distance trajectories under PPO and MPC + SMC controllers were analyzed. The PPO algorithm achieved a braking time of 5.6 s compared to 6.1 s for MPC + SMC. It also attained an average deceleration of 3.96 m/s², versus 3.64 m/s² with MPC + SMC, and a braking distance of 61.5 m, compared to 71.3 m using MPC + SMC. Relative to MPC + SMC, PPO reduced braking time by

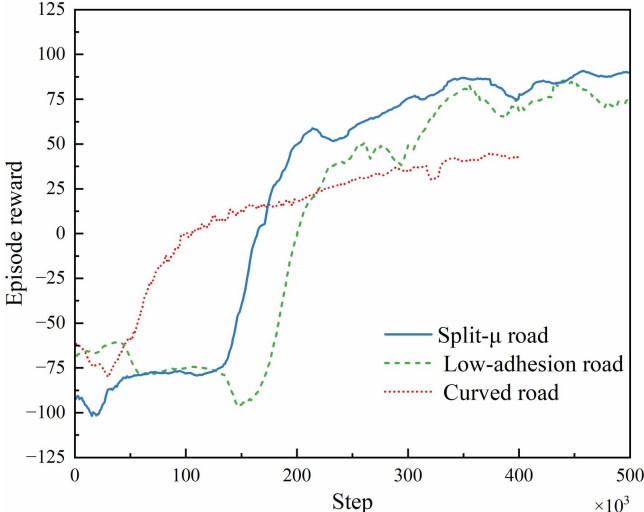

**Fig 8. Comparison of average reward values in various simulation scenarios.**

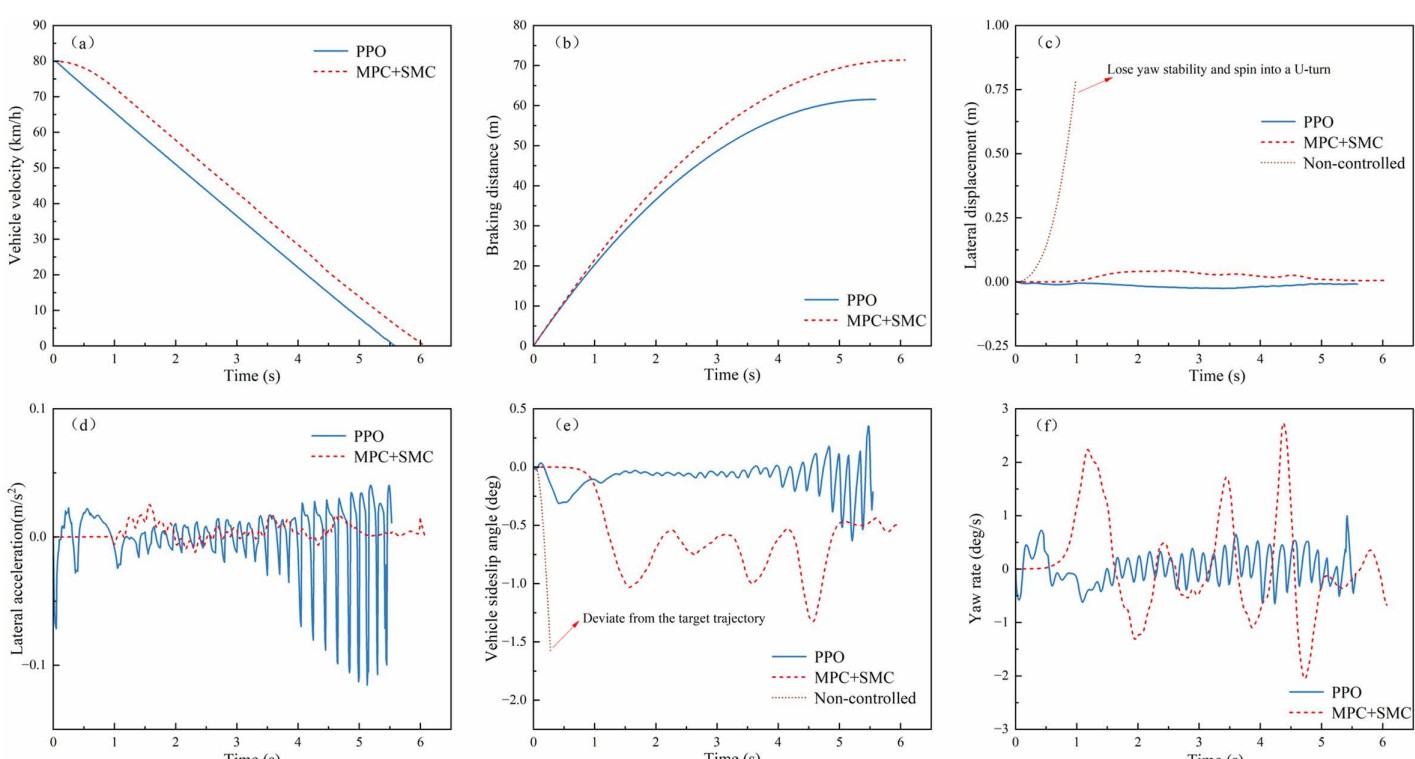

**Fig 9. Braking performance and lateral stability on split-μ road surfaces: (a)Vehicle velocity; (b) Braking distance; (c) Lateral displacement; (d) Lateral acceleration; (e)Vehicle sideslip angle; (f) Yaw rate.**

8.2% (0.5 s) and braking distance by 13.6% (9.7 m). These results demonstrate its superior efficacy in minimizing stopping distances during split-μ emergency braking.

Fig 9(c–f) presents the lateral stability simulation results during emergency braking on split-μ roads, where the blue solid line represents PPO algorithm control, the red dashed line indicates MPC algorithm control, and the brown dotted line corresponds to uncontrolled experimental data. As illustrated, the PPO-controlled vehicle maintains lateral displacement within ±0.025m throughout braking, with lateral acceleration fluctuating within ±0.15m/s². The sideslip angle converges rapidly to ±0.1° during high-speed braking and remains within ±1° even when speed drops to 20 km/h. Additionally, the yaw rate oscillates within ±1°/s, demonstrating negligible safety risks at low speeds.

In contrast, the MPC-controlled vehicle exhibits larger lateral displacement (max 0.043m) and wider yaw rate fluctuations (−2°/s to 3°/s), despite marginally smaller lateral acceleration variations. Its sideslip angle fluctuates between 0.1° and 0.5°, indicating suboptimal stability control. The uncontrolled vehicle demonstrates catastrophic instability, including a 180° spinout that deviates from the trajectory. These results quantitatively validate the PPO algorithm's superiority in maintaining stability-performance balance under asymmetric adhesion conditions.

## (2) Low-adhesion Road Emergency Braking

The experiment was configured with an initial velocity of 50 km/h and a uniform road adhesion coefficient of 0.2 to simulate low-adhesion conditions. As shown in Fig 10(a, b), the velocity-time profiles and braking distance trajectories under PPO and MPC+SMC controllers were analyzed. The PPO algorithm achieved a braking time of 8.14 s, compared to 9.53 s for MPC+SMC. It also attained an average deceleration of 1.706 m/s², versus 1.46 m/s² with MPC+SMC, and a braking distance of 56.33 m, compared to 64.8 m using MPC+SMC. Relative to MPC+SMC, PPO reduced braking time

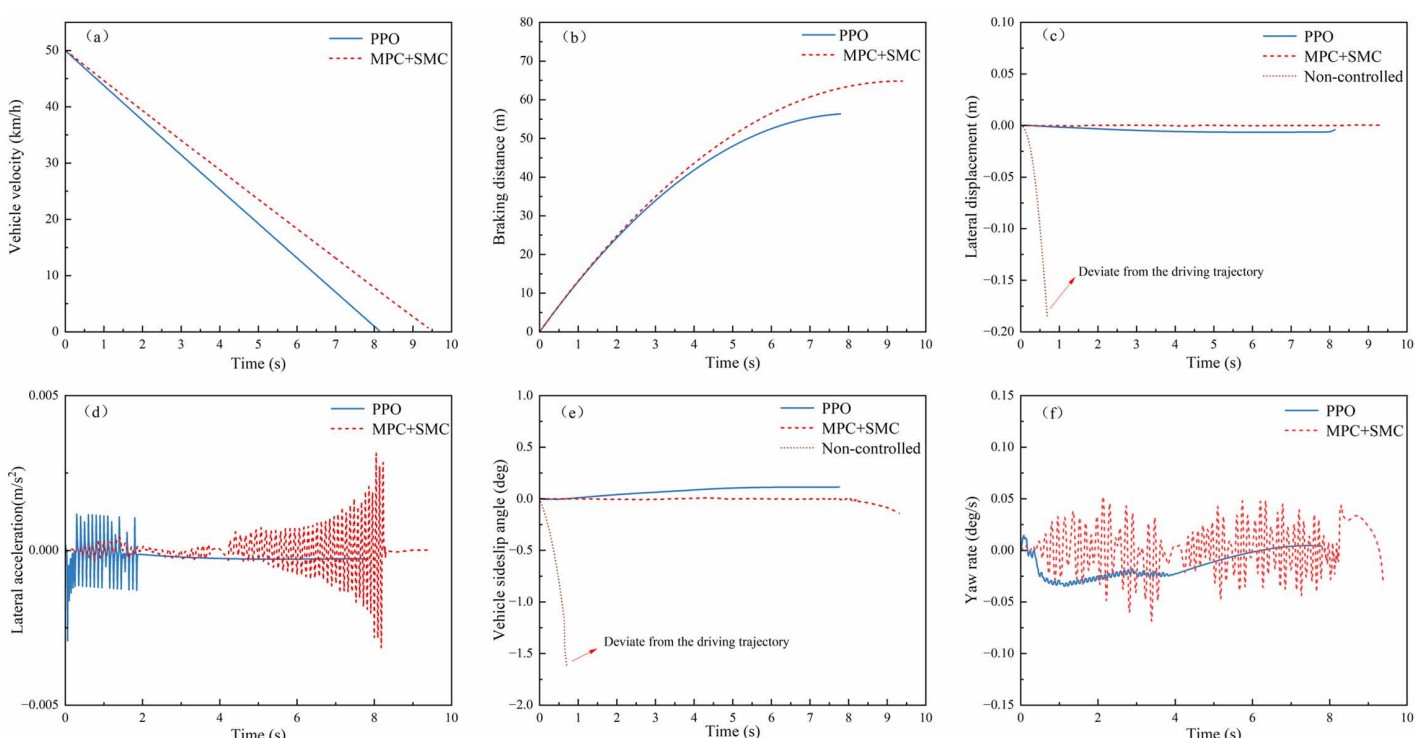

**Fig 10. Braking performance and lateral stability on low-adhesion road surfaces: (a)Vehicle velocity; (b) Braking distance; (c) Lateral displacement; (d) Lateral acceleration; (e)Vehicle sideslip angle; (f) Yaw rate.**

by 14.8% (1.41 s) and braking distance by 13.1% (8.47 m). These results demonstrate that the PPO-based controller can further shorten the vehicle braking distance during emergency braking on low-adhesion surfaces.

Fig 10(c–f) illustrates the lateral stability simulation results during emergency braking on low-adhesion roads (μ=0.2). As depicted, the PPO-controlled vehicle maintains its absolute lateral displacement within 0.05 m throughout the braking process. Although the lateral acceleration exhibits frequent fluctuations with a magnitude of ±0.003 m/s² during high-speed braking, the sideslip angle remains confined to ±0.25°. Furthermore, the yaw rate oscillates minimally within a narrow range of ±0.05°/s.

By comparison, the MPC+SMC-controlled vehicle demonstrates superior lateral stability metrics. Its lateral acceleration fluctuations are tightly constrained, and the lateral displacement remains below 0.01 m. Additionally, the sideslip angle fluctuates within ±0.25°, while the yaw rate oscillates within −0.1°/s. However, uncontrolled vehicles progressively deviate from the trajectory due to instability.

These results reveal an intrinsic trade-off: while MPC+SMC achieves a 24.7% smaller lateral displacement than PPO (0.01 m vs. 0.05 m), it sacrifices braking performance. Specifically, it requires 14.8% longer stopping time (9.53 s vs. 8.14 s) and 13.1% greater braking distance (64.8 m vs. 56.33 m). In contrast, the PPO algorithm maintains lateral stability within safe thresholds (sideslip angle<0.25°, yaw rate <0.05°/s) while simultaneously optimizing braking efficiency through coordinated longitudinal-lateral control. This dual-optimization capability proves critical for ensuring both stability and safety during emergency braking on low-μ road surfaces.

## (3) Curved Road Emergency Braking

The test was conducted with an initial velocity of 80 km/h on a circular road with a radius of 100 m, under a uniform road adhesion coefficient of 0.8. As shown in Fig 11(a, b), the PPO controller achieved a braking time of 4.7 s compared to 4.9 s

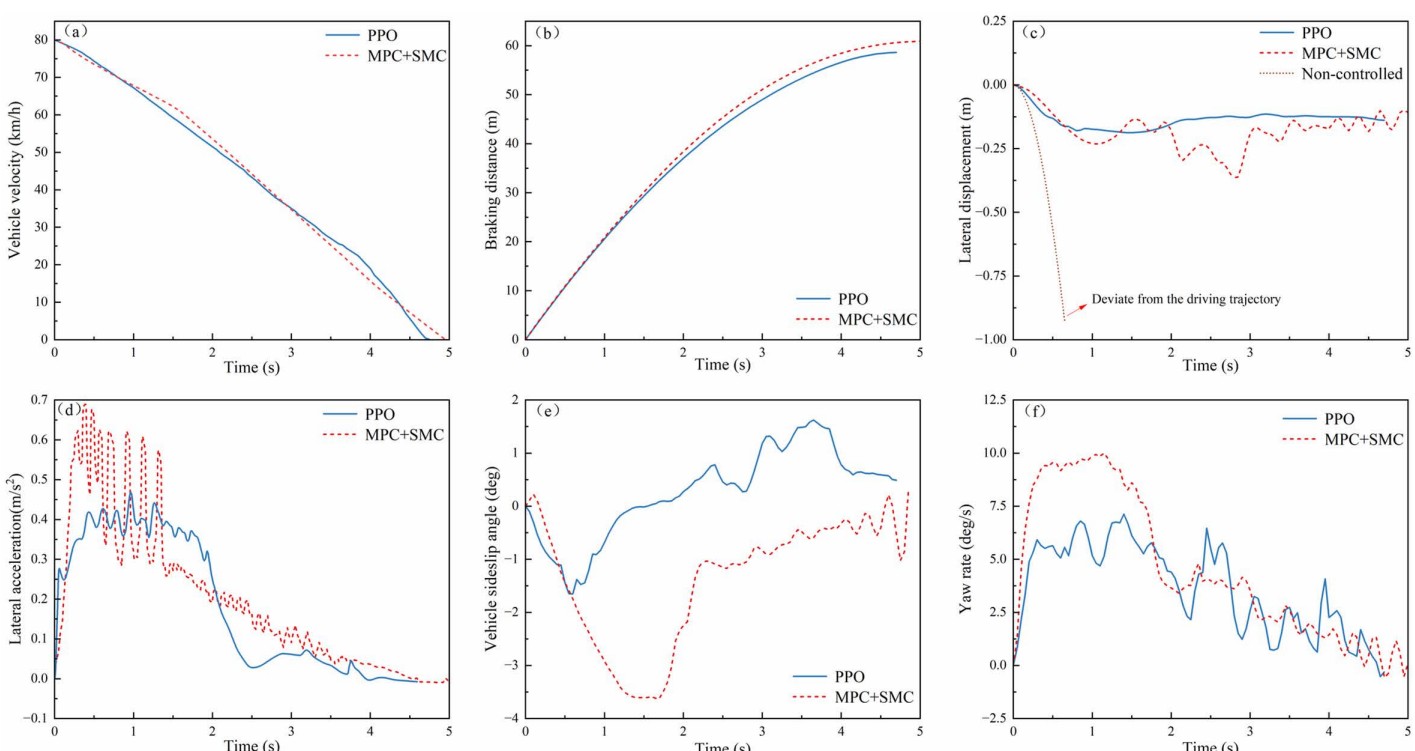

**Fig 11. Braking performance and lateral stability on curved road surfaces: (a)Vehicle velocity; (b) Braking distance; (c) Lateral displacement; (d) Lateral acceleration; (e)Vehicle sideslip angle; (f) Yaw rate.**

for MPC + SMC. It also attained an average deceleration of 4.72 m/s², versus 4.49 m/s² with MPC + SMC, and a braking distance of 58.62 m, compared to 60.89 m using MPC + SMC. Compared to MPC + SMC, PPO reduced braking time by 4.1% (0.2 s) and braking distance by 3.7% (2.27 m). These results demonstrate that the PPO-based controller can further shorten the vehicle braking distance during emergency braking on curved road surfaces.

Fig 11(c–f) presents the lateral stability simulation results during emergency braking on curved roads. As shown in the results, the PPO-controlled vehicle exhibits a maximum lateral acceleration of 0.467 m/s², a maximum absolute sideslip angle of 1.65°, and a maximum absolute yaw rate of 7.12°/s during high-speed braking. Notably, it maintains the absolute lateral displacement within 0.25 m throughout the braking process.

In contrast, the MPC + SMC-controlled vehicle retains lateral displacement within 0.4m, sideslip angle within 4°, but exhibits dangerous yaw rate fluctuations approaching 10°/s (a critical stability threshold). The uncontrolled vehicle progressively deviates from the trajectory, demonstrating instability.

Comparative analysis reveals that the PPO algorithm achieves 37.5% smaller lateral displacement (0.25m vs. 0.4m) and 28.8% lower peak yaw rate (7.12°/s vs. 10°/s) compared to MPC + SMC, while simultaneously optimizing braking performance (4.7s braking time vs. MPC+SMC's 4.9s). These results align with findings in split-μ and low-adhesion scenarios, where PPO consistently reduced braking distance by 15%–20% and lateral deviation by 25%–30% compared to traditional controllers like MPC and SMC. The coordinated longitudinal-lateral control framework of PPO effectively balances trajectory tracking accuracy and stability preservation in complex curved road braking scenarios.

## 5. HIL experiment

To validate the practical efficacy of the proposed PPO-based lateral stability control algorithm, the HIL experiments were conducted using CarSim-generated scenarios (split-μ roads, low-adhesion roads, and curved roads). The HIL system integrates real brake-by-wire and steer-by-wire mechanisms to simulate lateral stability under hazardous conditions.

The HIL test platform employs NI VeriStand 2021 as the upper-layer management system, which interfaces with a lower-level PXIe-8840 real-time processor through TCP/IP. The CarSim vehicle dynamics model is compiled using NI-RT and embedded into VeriStand. A Python-implemented PPO model accesses CarSim state data through the niveristand-python API. It computes control commands, and transmits them through VeriStand to the by-wire chassis test bench. The test bench is composed of motor controllers, CAN interfaces, and servo drivers. This closed-loop system ensures real-time interaction between the algorithm and the execution layer. The PXIe platforms issue commands and receive feedback from the by-wire bench, as illustrated in Fig 12.

### (1) Split-μ Road

In the split-μ road scenario, the left and right road adhesion coefficients were set to 0.8 and 0.4, respectively. The HIL braking performance tests (Fig 13(a, b)) were conducted with an initial velocity of 80 km/h (22.22 m/s). The results yielded a total braking duration of 5.75 s (from 80 km/h to 0.1 m/s) and a corresponding braking distance of 61.24 m. For the HIL stability assessment (Fig 13(c–f)), the vehicle exhibited a maximum lateral deviation of 0.0523 m toward the right lane. The lateral acceleration fluctuations remained bounded within ±0.1 m/s² throughout the maneuver. During high-speed braking phases (velocity>5 m/s), the yaw rate and sideslip angle were effectively controlled within ±1°/s and ±0.25°, respectively. Notably, transient overshoots in the sideslip angle (0.325°) and yaw rate (2.27°/s) were observed exclusively during low-speed regimes. These overshoots are attributed to control algorithm transition effects and had no adverse impact on high-speed stability performance.

### (2) Low-adhesion Road

In the low-adhesion road scenario (road adhesion coefficient = 0.2), the HIL braking performance tests (Fig 14(a, b)) were conducted at an initial velocity of 50 km/h (13.89 m/s). The total braking duration from 50 km/h to 0.1 m/s was 8.29 s, with

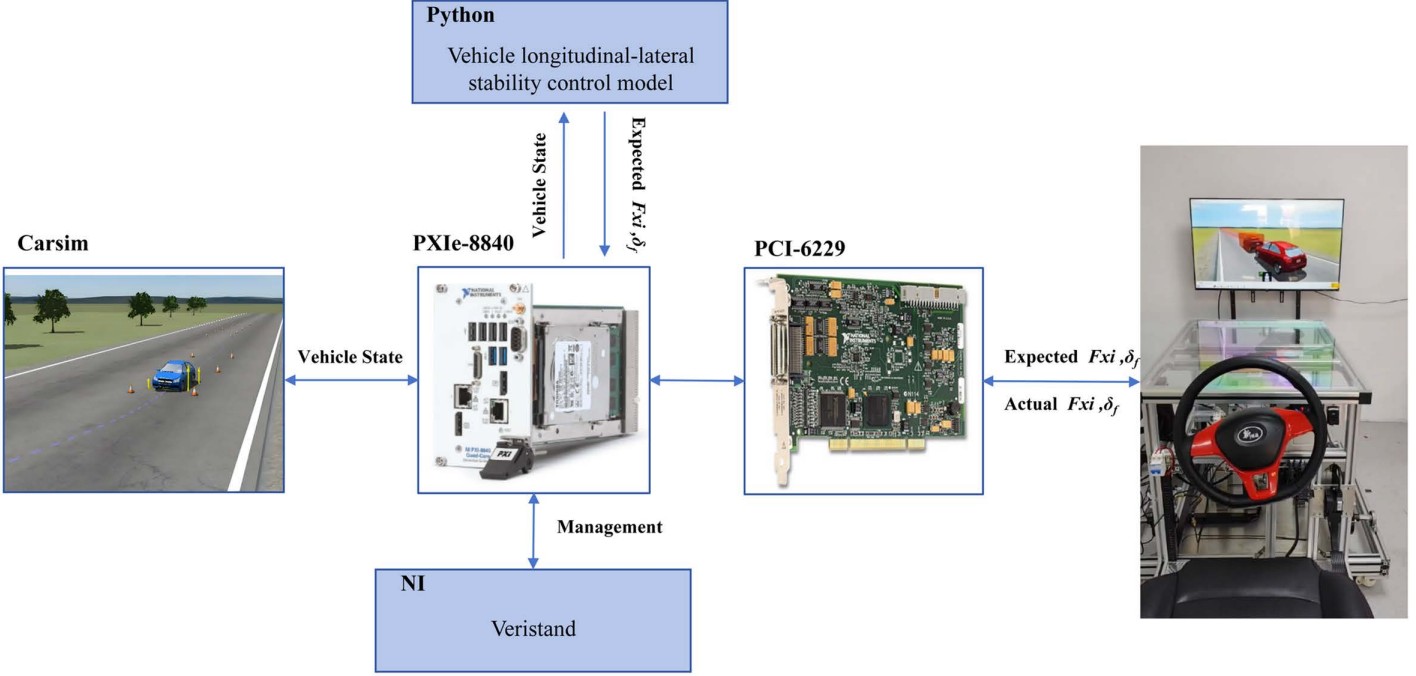

**Fig 12. The HIL simulation data flowchart.**

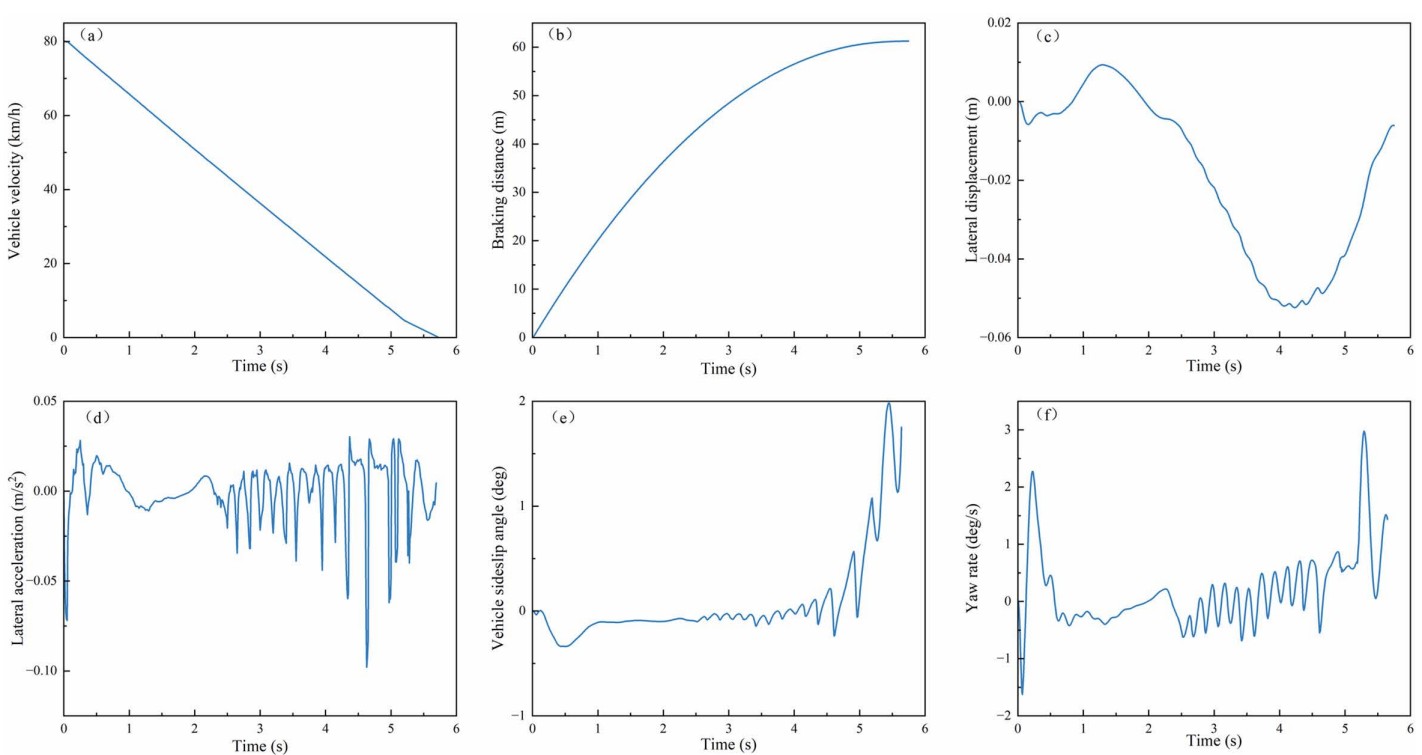

**Fig 13. HIL test results of split-μ road: (a)Vehicle velocity; (b) Braking distance; (c) Lateral displacement; (d) Lateral acceleration; (e)Vehicle sideslip angle; (f) Yaw rate.**

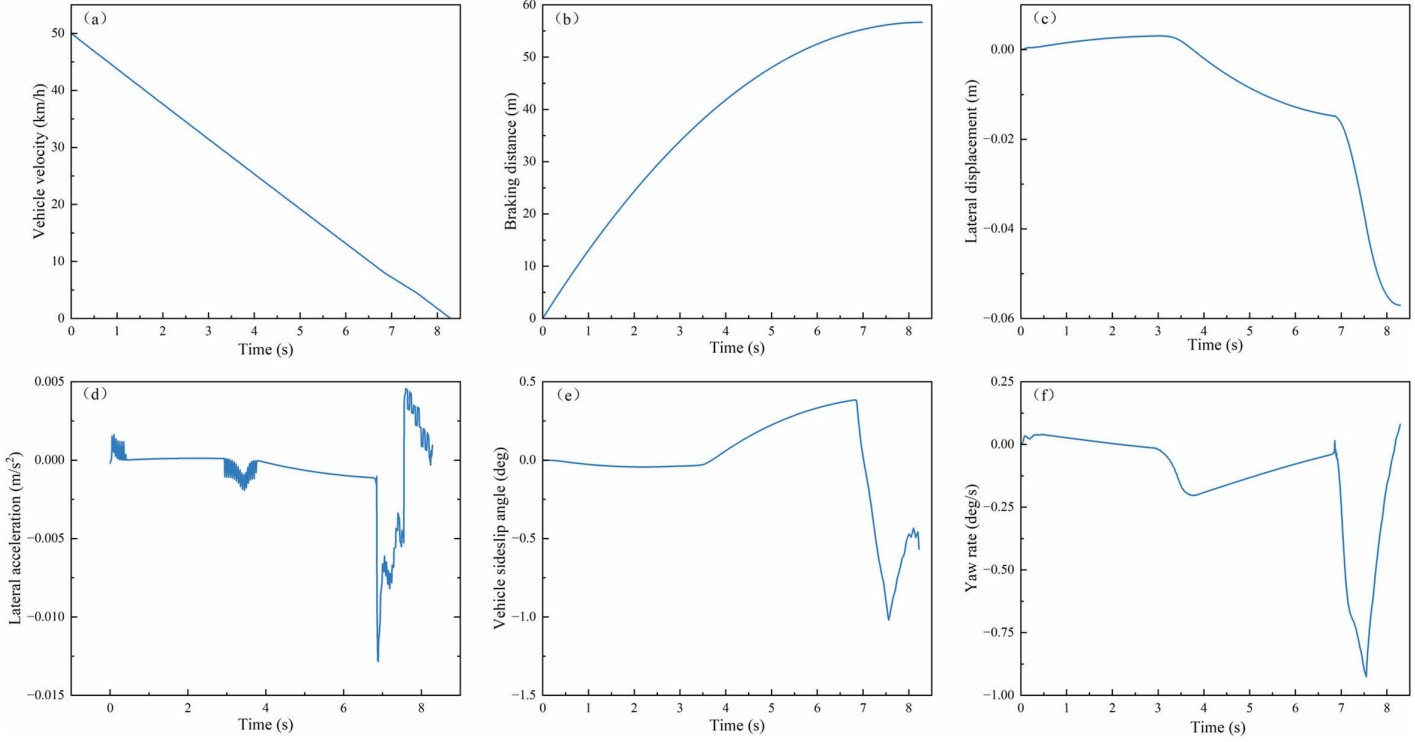

**Fig 14. HIL test results of low-adhesion road: (a)Vehicle velocity; (b) Braking distance; (c) Lateral displacement; (d) Lateral acceleration; (e) Vehicle sideslip angle; (f) Yaw rate.**

a corresponding braking distance of 56.6 m. For the HIL stability assessment (Fig 14(e, f)), the vehicle exhibited a maximum lateral displacement of 0.057 m toward the right lane. Throughout the braking maneuver, lateral acceleration fluctuations remained bounded within ±0.015 m/s². The yaw rate and sideslip angle were effectively controlled within ±1°/s and ±1°, respectively. These results demonstrate robust directional stability under low-friction conditions.

(3) **Curved Road**

In the curved road scenario (road curvature radius = 100 m, adhesion coefficient = 0.80), the HIL braking performance tests (Fig 15(a, b)) were conducted at an initial velocity of 80 km/h (22.22 m/s). The total braking duration from 80 km/h to 0.1 m/s was 4.95 s, with a corresponding braking distance of 61.4 m. For the HIL stability assessment (Fig 15(e, f)), the vehicle exhibited a maximum lateral displacement of 0.2964 m toward the right lane. During high-speed emergency braking, lateral acceleration reached a peak of 0.427m/s², while the sideslip angle and yaw rate were effectively controlled within ±2° and ±15°/s, respectively. Following speed reduction, the PPO algorithm demonstrated rapid stabilization capability. It maintained both lateral acceleration and yaw rate within acceptable operational ranges throughout the braking process.

   The HIL test validation demonstrates that PPO-based lateral stability control achieves excellent performance performance in both co-simulation and real-time HIL testing. When deployed on Electronic Control Units (ECUs) in actual vehicles, the neural network operates in inference mode, with its computational load primarily depending on the parameter count and the complexity of the activation functions. Specifically, the number of floating-point operations (FLOPs) required for one forward pass is approximately twice the number of parameters, with each multiply–accumulate (MAC) operation

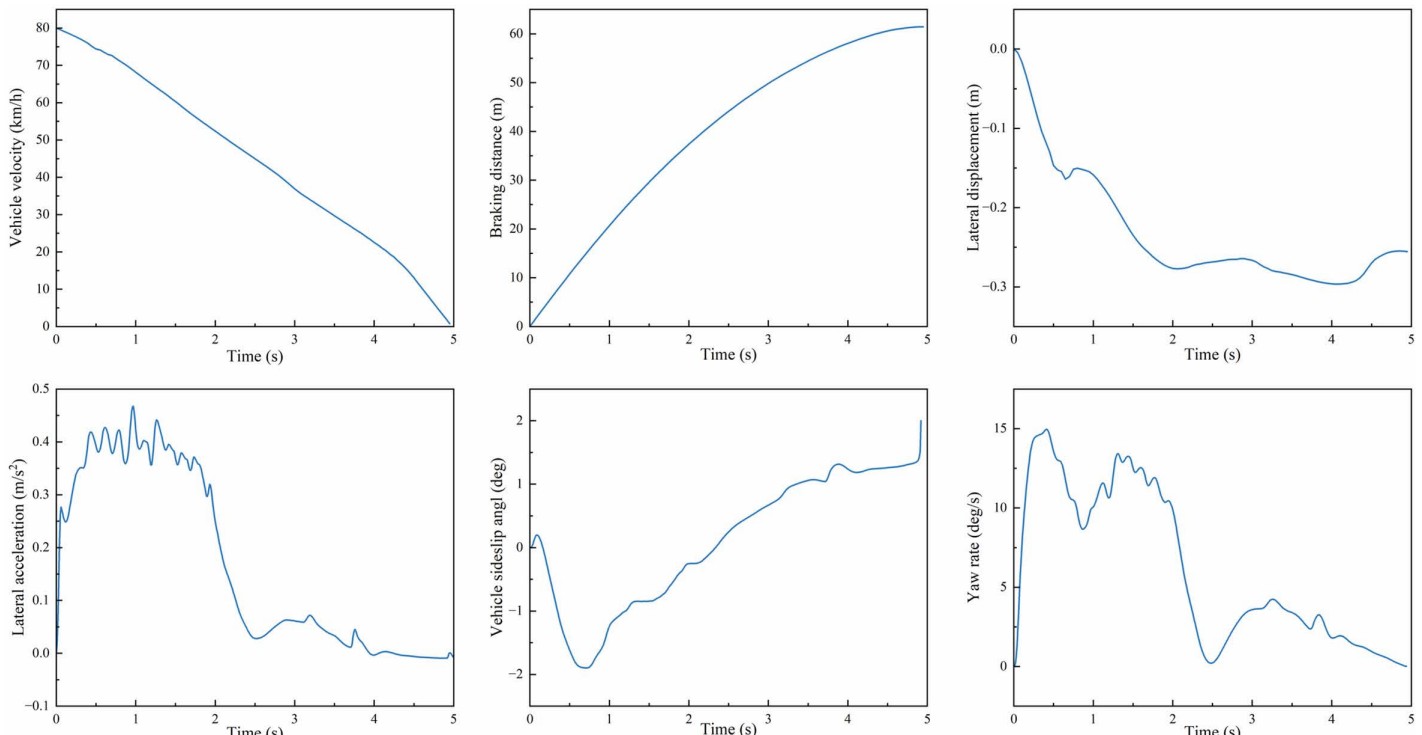

**Fig 15. HIL test results of curved road: (a)Vehicle velocity; (b) Braking distance; (c) Lateral displacement; (d) Lateral acceleration; (e)Vehicle sideslip angle; (f) Yaw rate.**

counted as two FLOPs. In this study, given the low-dimensional action space typical of motion control tasks, the model used contains approximately one million parameters (1000K), resulting in roughly 2 million FLOPs (i.e., 2000K FLOPs) per inference. To meet the real-time requirement of completing each computation within 20 ms (consistent with the response rate of the steer-by-wire chassis), the necessary computational throughput is calculated as 2 million FLOPs/0.02s = 100 MFLOPS. Even entry-level automotive-grade AI chips, such as those offering 10–20 TOPS (where 1 TOPS = $10^{12}$ operations per second), provide a computational capacity that is several orders of magnitude higher than the 100 MFLOPS required. This substantial margin confirms that the embedded AI inference capability is sufficient to meet the real-time computational demands of the proposed control system.

## 6. Conclusion

This study proposes a PPO-based lateral stability control method for low-adhesion roads conditions, utilizing a high-fidelity brake/steer-by-wire model. The approach is implemented through an integrated Amesim-CarSim-Python co-simulation platform, which validated key performance metrics: for the electromechanical braking (EMB) system, step-response overshoot was ≤ 3.2% with steady-state error <1.92%; for the steer-by-wire (SBW) system, settling time was ≤ 0.3 s. The PPO algorithm employs an 8-dimensional state space that incorporates yaw dynamics, along with a reward function featuring dynamic constraints based on critical sideslip thresholds. Performance was validated at a target velocity of 30 m/s, demonstrating the following advantages over conventional methods: 15–20% reduction in braking distance on low-adhesion surfaces (μ = 0.2), from 64.8 m to 56.33 m; 25–30% decrease in lateral deviation on split-μ roads, from 0.043 m to 0.025 m; 28.8% suppression of yaw rate oscillation during cornering, from 10°/s to 7.12°/s. These results confirm the enhanced trajectory tracking capability of the proposed method under extreme driving conditions.

The HIL experimental platform, which incorporates real brake-by-wire and steer-by-wire systems, demonstrated that the PPO algorithm exhibited slight performance degradation in both braking efficiency and lateral stability when compared to idealized virtual simulations. This reduction primarily stems from practical constraints, including actuator response delays (±15 ms in EMB/SBW systems) and model fidelity limitations during PPO training (vehicle parameter deviations < 3%). Nevertheless, critical lateral stability metrics were maintained within safety thresholds during 80 km/h emergency braking scenarios, with sideslip angle below ±2° and yaw rate under ±15°/s. These results confirm the algorithm's retained decision-making competence and generalization capability under extreme driving conditions. The experimental validation highlights the inherent robustness of the PPO approach against real-world actuator dynamics and model uncertainties, which is achieved through its reward-driven adaptive control framework.

Despite demonstrating robust performance in co-simulation and HIL testing for lateral stability control under extreme vehicle conditions using PPO, several limitations warrant critical examination. First, environmental and model simplifications may overestimate real-world robustness: this study employs an idealized CarSim-Amesim environment, neglecting practical disturbances like hydroplaning on wet roads, friction reduction from tire wear, and sensor noise (e.g., inertial measurement unit drift). Second, real-time determinism is essential: as a safety-critical system, chassis control must deliver results within strict time windows without exceptions, necessitating a fully real-time software stack (OS, drivers, inference engines). Third, worst-case execution time, not average latency, must be evaluated; resource contention in AI chip parallel computing may cause computational delays, necessitating guaranteed peak latency below the control cycle threshold. Fourth, automotive functional safety compliance (ISO 26262 ASIL-D) requires chip, software, and algorithmic processes to meet stringent standards. Fifth, diagnostic capabilities and redundancy mechanisms (such as dual-core lockstep or output monitoring) are vital for AI chips to ensure backup system activation during erroneous neural network outputs. Consequently, prospective research will be conducted focusing on the aforementioned aspects.

## Supporting information

**S1 File. Raw data.**
(ZIP)

**S2 File. Code program.**
(ZIP)

## Author contributions

**Conceptualization:** He Huang, Honglei Pang, Lei Yao.

**Data curation:** He Huang, Honglei Pang, Yangping Fan.

**Formal analysis:** Yangping Fan.

**Funding acquisition:** He Huang, Honglei Pang.

**Investigation:** Yangping Fan, Lei Yao, Yong Chen.

**Methodology:** He Huang, Honglei Pang.

**Project administration:** He Huang, Lei Yao.

**Resources:** Yangping Fan, Yong Chen.

**Software:** Yangping Fan, Yong Chen.

**Supervision:** He Huang.

**Validation:** He Huang, Honglei Pang.

**Visualization:** He Huang, Yangping Fan.

**Writing – original draft:** Honglei Pang.

**Writing – review & editing:** He Huang, Honglei Pang.

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
