## [Decision Letter · Decision Letter 0]

4 Sep 2025

PONE-D-25-42907Research on vehicle lateral stability control under low-adhesion road conditions using proximal policy optimization algorithmPLOS ONE

Dear Dr. Huang

Thank you for submitting your manuscript to PLOS ONE. After careful consideration, we feel that it has merit but does not fully meet PLOS ONE’s publication criteria as it currently stands. Therefore, we invite you to submit a revised version of the manuscript that addresses the points raised during the review process.

We look forward to receiving your revised manuscript.

Kind regards,

Jinhao Liang

Academic Editor

PLOS ONE

Journal Requirements:

“This study was supported by the Start-up Fund for New Talented Researchers of Nanjing Vocational University of Industry Technology(No.YK-22-04-03), Yangtze River Delta Sci-Tech Innovation Community Joint Research Project(2023CSJGG1600).”

**Additional Editor Comments:**

Please revise the manuscript thoroughly according to the reviewers’ comments. In addition, the literature review on vehicle lateral stability control should incorporate more recent studies, such as "A Robust Dynamic Game-Based Control Framework for Integrated Torque Vectoring and Active Front-Wheel Steering System, IEEE Transactions on Intelligent Transportation Systems, vol. 24, no. 7, pp. 7328-7341, July 2023", and "A Direct Yaw Moment Control Framework Through Robust T-S Fuzzy Approach Considering Vehicle Stability Margin, IEEE/ASME Transactions on Mechatronics, vol. 29, no. 1, pp. 166-178, Feb. 2024".

Reviewers' comments:

Reviewer's Responses to Questions

**Comments to the Author**

1. Is the manuscript technically sound, and do the data support the conclusions?

Reviewer #1: Yes

Reviewer #2: Yes

2. Has the statistical analysis been performed appropriately and rigorously? 

Reviewer #1: Yes

Reviewer #2: Yes

3. Have the authors made all data underlying the findings in their manuscript fully available?

Reviewer #1: Yes

Reviewer #2: Yes

4. Is the manuscript presented in an intelligible fashion and written in standard English?

Reviewer #1: Yes

Reviewer #2: Yes

5. Review Comments to the Author

Reviewer #1: This paper proposes to use PPO deep reinforcement learning to achieve longitudinal and horizontal cooperative stability control of low-attached pavement, and verifies it by Amesim-CarSim-Python co-simulation and HIL experiments. The overall idea is clear, the experiment is sufficient, and the results are credible. However, there are still some problems, it is recommended to modify and adopt, see the following for the specific problems:

1.The limitations of deep reinforcement learning (DRL) (safety, interpretability, training costs) are not mentioned. The authors should explain these limitations in a few sentences.

2.There is no uniform definition of “low-adhesion road surface.” Line 122 of the manuscript states that μ < 0.3, but line 596 states that μ = 0.2. It is recommended that the definition of “low friction” be standardized.

3.The symbols ε1 and ε2 in formula (13) have no values or references. The author needs to provide reference values or explanations.

4.The parameters in Table 4 do not have physical dimensions. The author should explain this to prevent misunderstanding by readers.

5.Table 4 ω₁=-2 may make the reward value negative, and its physical meaning needs to be explained, does this punish excessive braking?

6.The PPO does not specify the network structure, activation function, observation normalization method, random seed, or training-testing data division ratio, making it difficult to reproduce. The authors should supplement the table with an explanation of the number of network layers, number of nodes, and training-testing data division ratio.

7.Reference 26 contains an error in the DOI number (2000-01-1060 corresponds to 2001-01-1060).

8.The overall clarity of the picture is not high enough, and the resolution of each schematic needs to be improved.

Reviewer #2: This manuscript presents a study on an integrated longitudinal and lateral vehicle stability control system using the Proximal Policy Optimization (PPO) deep reinforcement learning algorithm. The simulation experiments verified the effectiveness of the proposed algorithm. Some issues need to be resolved before the manuscript can be accepted.

1. A more detailed description of the hyperparameter tuning methodology is recommended.

2. The manuscript currently lacks sufficient comparisons with other learning algorithms to highlight the superiority of the proposed method.

3. It is recommended that the authors conduct a thorough discussion on the limitations of the current research and provide an evaluation of the feasibility of deploying it on real automotive electronic control units.

4. The problem of vehicle lateral control is a well-established example of motion control for systems under nonholonomic constraints, a field with extensive research. It is suggested that the introduction incorporate more key studies on the lateral control of nonholonomic systems. Doing so would not only enrich the literature review but also help to define the unique contributions of this study and underscore the significance of tackling this classic problem with advanced reinforcement learning techniques more sharply.

a) "Paired Interactions of Magnetic Millirobots in Confined Spaces Through Data-Driven Disturbance Rejection Control Under Global Input," IEEE/ASME Transactions on Mechatronics, doi: 10.1109/TMECH.2024.3521085.

b) "A Robot Motion Learning Method Using Broad Learning System Verified by Small-Scale Fish-Like Robot," in IEEE Transactions on Cybernetics, vol. 53, no. 9, pp. 6053-6065, Sept. 2023, doi: 10.1109/TCYB.2023.3269773.

5. The manuscript is generally well-written, but a few minor grammatical checks and rephrasing could improve readability. For example, some sentences are quite long and could be broken down.

6. The manuscript’s images are blurry and high-resolution versions should be provided.

6. PLOS authors have the option to publish the peer review history of their article (what does this mean? ). If published, this will include your full peer review and any attached files.

**Do you want your identity to be public for this peer review?** For information about this choice, including consent withdrawal, please see our Privacy Policy .

Reviewer #1: No

Reviewer #2: No

---

## [Author Response · Author response to Decision Letter 1]

12 Oct 2025

Dear Editor We hereby express our sincere gratitude to the academic editor and reviewers for their valuable comments. We have made corresponding revisions to the manuscript in accordance with the comments from the academic editor and reviewers, and have provided point-by-point responses to each comment raised by them.

---

## [Decision Letter · Decision Letter 1]

15 Oct 2025

Research on vehicle lateral stability control under low-adhesion road conditions using proximal policy optimization algorithm

PONE-D-25-42907R1

Dear Dr. Huang,

We’re pleased to inform you that your manuscript has been judged scientifically suitable for publication and will be formally accepted for publication once it meets all outstanding technical requirements.

Kind regards,

Jinhao Liang

Academic Editor

PLOS ONE

Additional Editor Comments (optional):

Reviewers' comments:

Reviewer's Responses to Questions

**Comments to the Author**

1. If the authors have adequately addressed your comments raised in a previous round of review and you feel that this manuscript is now acceptable for publication, you may indicate that here to bypass the “Comments to the Author” section, enter your conflict of interest statement in the “Confidential to Editor” section, and submit your "Accept" recommendation.

Reviewer #2: All comments have been addressed

Reviewer #3: (No Response)

2. Is the manuscript technically sound, and do the data support the conclusions?

Reviewer #2: Yes

Reviewer #3: Yes

3. Has the statistical analysis been performed appropriately and rigorously? 

Reviewer #2: Yes

Reviewer #3: Yes

4. Have the authors made all data underlying the findings in their manuscript fully available?

Reviewer #2: Yes

Reviewer #3: Yes

5. Is the manuscript presented in an intelligible fashion and written in standard English?

Reviewer #2: Yes

Reviewer #3: Yes

6. Review Comments to the Author

Reviewer #2: (No Response)

Reviewer #3: The author proposed a research on vehicle lateral stability control under low-adhesion road conditions using proximal policy optimization algorithm, and provided detailed data analysis. This research is rich in content and the conclusions are reasonable.

7. PLOS authors have the option to publish the peer review history of their article (what does this mean? ). If published, this will include your full peer review and any attached files.

**Do you want your identity to be public for this peer review?** For information about this choice, including consent withdrawal, please see our Privacy Policy .

Reviewer #2: No

Reviewer #3: No

---

## [Editor Report · Acceptance letter]

PONE-D-25-42907R1

PLOS ONE

Dear Dr. Huang,

I'm pleased to inform you that your manuscript has been deemed suitable for publication in PLOS ONE. Congratulations! Your manuscript is now being handed over to our production team.

Kind regards,

on behalf of

Dr. Jinhao Liang

Academic Editor

PLOS ONE